**Subject Category:**
Biology (whole organism)

health and disease and epidemiology/behaviour/ molecular biology

biological age, telomere length, telomere attrition, smoking, longitudinal

**Author for correspondence:**
Melissa Bateson
e-mail: melissa.bateson@ncl.ac.uk

[†]J.M.S. died while this paper was under review and the final version was prepared posthumously.

# Smoking does not accelerate leucocyte telomere attrition: a meta-analysis of 18 longitudinal cohorts

Melissa Bateson[1], Abraham Aviv[2], Laila Bendix[3], Athanase Benetos[4], Yoav Ben-Shlomo[6], Stig E. Bojesen[7], Cyrus Cooper[8], Rachel Cooper[9], Ian J. Deary[10], Sara Hägg[12], Sarah E. Harris[10,13], Jeremy D. Kark[14], Florian Kronenberg[15], Diana Kuh[9], Carlos Labat[5], Carmen M. Martin-Ruiz[1], Craig Meyer[17], Børge G. Nordestgaard[7], Brenda W. J. H. Penninx[18], Gillian V. Pepper[1], Dóra Révész[19], M. Abdullah Said[20], John M. Starr[10,11,†], Holly Syddall[8], William Murray Thomson[21], Pim van der Harst[20], Mary Whooley[17], Thomas von Zglinicki[22,23], Peter Willeit[16,24], Yiqiang Zhan[12] and Daniel Nettle[1]

[1]Centre for Behaviour and Evolution and Institute of Neuroscience, Newcastle University, Newcastle upon Tyne NE2 4HH, UK
[2]Center of Human Development and Aging, New Jersey Medical School, Rutgers University, Newark, NJ 07103, USA
[3]Pain Center South, Department of Anesthesiology and Intensive Care Medicine, University Hospital Odense, Odense, Denmark
[4]Department of Geriatric Medicine, CHRU de Nancy, and [5]INSERM U1116, Université de Lorraine, Nancy, France
[6]School of Social and Community Medicine, University of Bristol, Canynge Hall, Bristol, UK
[7]Department of Clinical Biochemistry, Herlev and Gentofte Hospital, Copenhagen University Hospital, Copenhagen University, Copenhagen, 2730 Herlev, Denmark
[8]MRC Lifecourse Epidemiology Unit, University of Southampton, Southampton General Hospital, Southampton SO16 6YD, UK
[9]MRC Unit for Lifelong Health and Ageing at UCL, University College London, 33 Bedford Place, London WC1B 5JU, UK
[10]Centre for Cognitive Ageing and Cognitive Epidemiology, Department of Psychology, and [11]Alzheimer Scotland Dementia Research Centre, University of Edinburgh, Edinburgh EH8 9JZ, UK

[12]Department of Medical Epidemiology and Biostatistics, Karolinska Institutet, 17177 Stockholm, Sweden

[13]Medical Genetics Section, University of Edinburgh Centre for Genomic and Experimental Medicine and MRC Institute of Genetics and Molecular Medicine, Edinburgh EH4 2XU, UK

[14]Hebrew University–Hadassah School of Public Health and Community Medicine, Ein Kerem, Jerusalem, Israel

[15]Division of Genetic Epidemiology, Department of Medical Genetics, Molecular and Clinical Pharmacology, and [16]Department of Neurology, Medical University of Innsbruck, Innsbruck 6020, Austria

[17]Department of Medicine, University of California, San Francisco, CA 94121, USA

[18]Department of Psychiatry, VU University Medical Center, Oldenaller 1, 1081 HJ Amsterdam, The Netherlands

[19]Department of Epidemiology, GROW School for Oncology and Developmental Biology, Maastricht University, PO Box 616, 6200 MD Maastricht, The Netherlands

[20]Department of Cardiology, University Medical Center Groningen, University of Groningen, Groningen 9700 RB, The Netherlands

[21]Sir John Walsh Research Institute, Faculty of Dentistry, University of Otago, Dunedin 9054, New Zealand

[22]Institute for Cell and Molecular Biosciences, Newcastle University, Newcastle upon Tyne, UK

[23]Arts and Sciences Faculty, Molecular Biology and Genetics, Near East University, Nicosia, North Cyprus, Mersin 10, Turkey

[24]Department of Public Health and Primary Care, University of Cambridge, Cambridge, UK

MB, 0000-0002-0861-0191; AA, 0000-0002-7441-0227; DN, 0000-0001-9089-2599

Smoking is associated with shorter leucocyte telomere length (LTL), a biomarker of increased morbidity and reduced longevity. This association is widely interpreted as evidence that smoking causes accelerated LTL attrition in adulthood, but the evidence for this is inconsistent. We analysed the association between smoking and LTL dynamics in 18 longitudinal cohorts. The dataset included data from 12 579 adults (4678 current smokers and 7901 non-smokers) over a mean follow-up interval of 8.6 years. Meta-analysis confirmed a cross-sectional difference in LTL between smokers and non-smokers, with mean LTL 84.61 bp shorter in smokers (95% CI: 22.62 to 146.61). However, LTL attrition was only 0.51 bp yr$^{-1}$ faster in smokers than in non-smokers (95% CI: −2.09 to 1.08), a difference that equates to only 1.32% of the estimated age-related loss of 38.33 bp yr$^{-1}$. Assuming a linear effect of smoking, 167 years of smoking would be required to generate the observed cross-sectional difference in LTL. Therefore, the difference in LTL between smokers and non-smokers is extremely unlikely to be explained by a linear, causal effect of smoking. Selective adoption, whereby individuals with short telomeres are more likely to start smoking, needs to be considered as a more plausible explanation for the observed pattern of telomere dynamics.

# 1. Introduction

Leucocyte telomere length (LTL)—the length of the repeated TTAGGG sequence at the end of leucocyte chromosomes—is an extensively studied biomarker of human health and well-being. In support of a link between poorer health and shorter average LTL, many cross-sectional studies have found an association between tobacco smoking and shorter LTL [1,2]. Since human telomeres shorten with age, these data are widely interpreted as demonstrating that smoking accelerates the rate of biological ageing [1,3,4]. For example, Valdes *et al.* [3] concluded, 'Our findings suggest that obesity and cigarette smoking accelerate human ageing . . . smoking a pack a day for 40 years corresponds to 7.4 years of ageing'. Thus, the telomere data are invoked to support a more general claim that smoking is a potent gerontogen, or ageing accelerator [5,6].

Smoking undoubtedly has a myriad of negative effects on human health and longevity. Moreover, the hypothesis that smoking causes telomere attrition is mechanistically plausible. Smoking causes increased levels of oxidative stress and inflammation [7–9], both of which are implicated in telomere attrition [10]. *In vitro* studies show that oxidative stress increases telomere attrition by increasing telomere loss per cell replication in a dose-dependent manner [11,12]. Smoking is therefore assumed to accelerate the rate of telomere attrition. Thus, although correlation does not provide evidence for causation, the hypothesis that smoking causes accelerated telomere attrition *in vivo* has been uncritically accepted based on cross-sectional data, and alternative explanations for the observed association between smoking and LTL have not been considered. However, evidence has started to accumulate that challenges this view. First, predicted links between oxidative stress and telomere attrition *in vivo* are proving elusive [13]. Second, a Mendelian randomization study that used a genetic polymorphism (*CHRNA3* genotype) established to be strongly associated with tobacco consumption

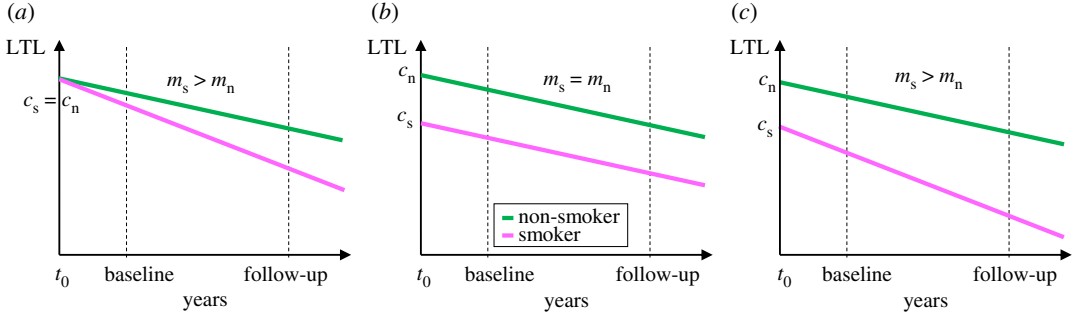

**Figure 1.** Three alternative models to explain the observed association between smoking and LTL: (a) causation, (b) selective adoption, and (c) mixed (=causation + selective adoption). We assume that smoking starts at $t_0$ for smokers and continues thereafter, whereas non-smokers never smoke. The magenta line represents the telomere dynamics for smokers and the green line for age-matched non-smokers. The dotted lines represent the position of two measurements of adult LTL (baseline and follow-up) made after the start of smoking.

found no evidence to support a causal association between smoking and short telomeres [14]. Finally, most of the variation in adult LTL is already present in early life, prior to the age at which most children start smoking, and adult LTL rank is largely stable [15,16]. Thus, there appears to be little scope for smoking to influence variation in adult LTL. Our aim in the current paper is therefore to use a meta-analysis of longitudinal LTL attrition data to directly test the hypothesis that smoking causes a sustained increase in the rate of LTL attrition in adults (henceforth the causation hypothesis). We additionally propose an alternative hypothesis of selective adoption, whereby individuals with shorter LTL are more likely to start smoking.

To generate testable predictions from the causation and selective adoption hypotheses we start by stating their assumptions (figure 1). For smokers, we assume that from the point at which an individual starts smoking, their LTL in subsequent years can be modelled as a straight line with a positive intercept ($c_s$) corresponding to their LTL at the start of smoking, and a negative slope ($m_s$) corresponding to their LTL attrition per year of smoking. For age-matched non-smokers, we also assume that LTL can be modelled as a straight line with a positive intercept and negative slope with values $c_n$ and $m_n$ respectively. The causation hypothesis assumes that prior to starting smoking there is no difference in the LTL of future smokers and non-smokers (i.e. $c_s = c_n$), but that after starting smoking the rate of telomere attrition is higher for smokers than for non-smokers ($m_s > m_n$). By contrast, selective adoption assumes that prior to starting smoking future smokers have shorter LTL than future non-smokers ($c_s < c_n$), but after starting smoking the rate of LTL attrition is equal in smokers and non-smokers ($m_s = m_n$). Since causation and selective adoption are not mutually exclusive, we also consider a mixed hypothesis that assumes that future smokers have shorter LTL than future non-smokers ($c_s < c_n$) and after starting smoking the rate of LTL attrition is faster in smokers than in non-smokers ($m_s > m_n$).

Two major predictions emerge from figure 1 that are tested here. First, all three hypotheses predict that LTL for smokers should be shorter than LTL for non-smokers at any time-point following the adoption of smoking. Thus, the observation of shorter LTL in adult smokers cannot be used to distinguish between the hypotheses. Second, the selective adoption hypothesis is unique in predicting that adult LTL attrition rate should be equal in current smokers and non-smokers, whereas the causation and mixed hypotheses both predict faster attrition in smokers. There have been a number of attempts to test this latter prediction using longitudinal LTL attrition data, but they have produced inconsistent findings [17]. Whereas some studies report faster LTL attrition in smokers [18,19], others report no difference [20–25] and one study reports faster LTL attrition in non-smokers [26]. To discriminate selective adoption from the other hypotheses it is critical to establish whether LTL attrition rates differ between smokers and non-smokers, and whether the observed difference in attrition is sufficient to explain the observed cross-sectional difference in LTL. Here we present the first quantitative meta-analysis aimed at estimating the effect of smoking on the rate of LTL attrition. This analysis was based on all published datasets that we were able to identify that contained relevant longitudinal measurements of LTL and information on smoking status, regardless of whether the association between smoking and LTL dynamics had been previously described.

# 2. Methods

## 2.1. Search strategy and data eligibility requirements

We sought longitudinal cohort studies in which baseline and follow-up measurements of LTL were obtained from each participant and in which data on smoking behaviour were also available. To allow sufficient time for biologically meaningful changes in LTL to be observed, we required a follow-up interval of at least 4 years [27].

Electronic supplementary material, figure S1 provides a PRISMA diagram summarizing how the data were obtained. We performed a systematic literature search using ISI Web of Knowledge (Thomson Scientific Technical Support, New York) for articles published prior to 2017 using the search terms: 'telomere*' and 'longitudinal' and 'smoking'. Additional relevant articles were identified by snowballing [28]. Articles were screened to identify potentially eligible cohorts. In many cases, determining the effect of smoking on LTL attrition was not the primary aim of an article, but smoking status was mentioned among the list of control variables.

In articles that reported estimates of the association between smoking and LTL attrition, the latter estimates usually came from multiple regression models that had been adjusted for baseline LTL. This practice is nearly universal due to the strong association present in most datasets between baseline LTL and rate of LTL attrition arising largely as a consequence of regression to the mean [29]. We have demonstrated that such adjusted estimates are likely to be biased in a direction that exaggerates the true effect of smoking on LTL attrition [30]. Therefore, we sought unadjusted estimates of the difference in LTL dynamics between smokers and non-smokers.

In two cases we were able to extract the required summary statistics from published articles, but for the majority of eligible cohorts, summary statistics on raw LTL and LTL attrition for smokers and non-smokers were not reported. Therefore, we contacted the authors and/or cohort managers to request either the required summary statistics, or the full individual participant-level data. In order to ensure consistency across cohorts in how the data were analysed, in the cases where we requested summary statistics, we provided detailed instructions on how these were to be calculated (including any exclusions—see below). In nine cases authors or cohort managers calculated and supplied the required summary statistics, and in the remaining seven cases the first author (M.B.) obtained raw data and calculated the summary statistics herself. For nine cohorts there were some published findings available relating to the effect of smoking on LTL attrition rate (for a narrative review see [17]), but for the other nine cohorts the first author was blind to the effect of smoking at the time of requesting the data (see table 1 for details).

The following data were obtained for each cohort included in the meta-analysis: the number of participants (smokers and non-smokers), mean age at baseline measurement (in years), mean follow-up interval (in years), LTL measurement method (TRF or qPCR), LTL measurement units (base pairs or T/S ratios), mean LTL at baseline and follow-up for smokers and non-smokers, standard deviation of LTL length at baseline and follow-up for smokers and non-smokers, annual rate of LTL attrition for smokers and non-smokers, standard deviation of LTL attrition per year for smokers and non-smokers, and the Pearson correlation between LTL at baseline and follow-up measurements.

Participants were only included if they had LTL data at both baseline and follow-up; participants lost to follow-up were excluded. In cohorts in which LTL was measured at more than two time-points (LBC1921 and LBC1936) we only used the two LTL measurements that gave the longest follow-up interval for each participant, designating the first as baseline and the second as follow-up. LTL attrition rate for a participant was calculated via the following formula: LTL attrition rate (bp yr$^{-1}$) = (LTL$_{baseline}$ − LTL$_{follow-up}$)/follow-up years.

We sought to only include data from participants who were either consistent never smokers or consistent current smokers at baseline and follow-up; where possible, we excluded participants who had quit smoking prior to baseline or between baseline and follow-up. We did not attempt to explore the effects of amount smoked, since consistent data were not available for all cohorts. For the majority of cohorts (15/18), smokers were defined as consistent current smokers at both baseline and follow-up and non-smokers were defined as consistent never-smokers at baseline and follow-up; individuals who changed smoking status between baseline and follow-up were excluded. However, for one cohort (BRUNECK), smokers and non-smokers were defined based on baseline status only, and for the two cohorts, for which data were extracted from published papers (BHS and ESTHER), it was not clear whether the smoker/non-smoker classification was based on status at baseline, follow-up or both.

**Table 1.** Details of the cohorts included.

| cohort (acronym) | country | mean age at baseline (years) | mean follow-up interval (years) | number of participants with consistent smoking status over the follow-up interval[a] | | source of TL data[b] | effect of smoking on LTL attrition published? | reference for cohort |
| --- | --- | --- | --- | --- | --- | --- | --- | --- |
| | | | | smokers | non-smokers | | | |
| ADELAHYDE (ADE) | France | 68.3 | 8.3 | 2 | 42 | raw | yes | [31] |
| Bogalusa Heart Study (BHS) | USA | 31.4 | 5.9 | 214 | 421 | extracted | no | [20] |
| Bruneck Study (BRUNECK) | Italy | 58.6 | 10.0 | 114 | 307 | summary | no | [21] |
| Copenhagen City Heart Study (CCHS) | Denmark | 52.7 | 9.4 | 1402 | 1455 | summary | no | [25] |
| Caerphilly Cohort Study (CCS) | Wales, UK | 64.4 | 8.0 | 106 | 154 | raw | yes | [32] |
| Dunedin Multidisciplinary Health and Development Study (DMHDS) | New Zealand | 26.0 | 12.0 | 173 | 458 | summary | yes | [33] |
| Evolution de la Rigidité Artérielle (ERA) | France | 59.8 | 9.5 | 1 | 86 | raw | no | [24] |
| Epidemiological Study on the Chances of Prevention, Early Recognition, and Optimised Treatment of Chronic Diseases in the Older Population (ESTHER) | Germany | 61.3 | 8.0 | 116 (631) | 483 (1696) | extracted | no | [26] |
| Hertfordshire Ageing Study (HAS) | England, UK | 67.1 | 9.3 | 20 | 87 | raw | yes | [32] |
| Heart and Soul Study (HSS) | USA | 66.1 | 4.8 | 76 | 475 | summary | no | [22] |
| Jerusalem LRC Study (JLRCS) | Israel | 30.1 | 13.1 | 154 | 275 | summary | yes | [34] |
| Lothian Birth Cohort 1921 (LBC1921) | Scotland, UK | 80.2 | 9.1 | 3 | 76 | raw | yes | [32] |
| Lothian Birth Cohort 1936 (LBC1936) | Scotland, UK | 69.6 | 6.0 | 62 | 406 | raw | yes | [35] |
| Danish MONICA1 and 10 survey (MONICA) | Denmark | 44.1 | 10.9 | 532 | 603 | summary | no | [18] |
| Netherlands Study of Depression and Anxiety (NESDA) | Netherlands | 40.8 | 6.0 | 455 | 489 | summary | no | [23] |
| MRC National Survey of Health and Development (NSHD) | UK | 53.4 | 9.3 | 112 | 326 | raw | yes | [32] |

(Continued.)

**Table 1.** (*Continued.*)

| cohort (acronym) | country | mean age at baseline (years) | mean follow-up interval (years) | number of participants with consistent smoking status over the follow-up interval[a] | | source of TL data[b] | effect of smoking on LTL attrition published? | reference for cohort |
|---|---|---|---|---|---|---|---|---|
| | | | | smokers | non-smokers | | | |
| Prevention of Renal and Vascular End-Stage Disease (PREVEND) | Netherlands | 46.6 | 6.5 | 1091 | 1456 | summary | no | [19] |
| Swedish Adoption/Twin Study of Aging (SATSA) | Sweden | 66.7 | 9.4 | 45 | 302 | summary | yes | [36] |

[a]These numbers are often smaller than the numbers given in the original reference for the cohort due to the fact that we only included participants with consistent smoking status between baseline and follow-up (see Methods for details). For ESTHER, analyses of baseline LTL are based on the numbers in brackets.

[b]Data sources: raw = raw TL data obtained from cohort manager and summary statistics calculated by first author (M.B.); extracted = summary statistics extracted from published article (reference in final column); summary = summary statistics calculated and supplied by co-authors (see Authors' contributions section for details).

## 2.2. Statistical analysis

We analysed the data in the statistical programming language R, v. 3.3.3 [37], using the meta-analysis package 'metafor' [38].

Since some cohorts reported LTL measurements in T/S ratios and others in base pairs, we computed standardized mean differences (SMDs) between smokers and non-smokers for LTL and LTL attrition. We used random-effects meta-analysis models throughout, because we deemed the cohorts too different to justify assuming that there are common true effect sizes to estimate [39]. Unless otherwise stated, we estimated parameters using inverse-variance weighting of cohorts (the default) and REML. In one of our sensitivity analyses we instead weighted cohorts by the correlation between baseline and follow-up LTL measurements ($r$). The rationale for this decision is that $r$ has been argued to be a good proxy for LTL measurement error [27,40], an assumption supported by the observation that datasets measured with more precise TRF always have higher correlations than those measured with less precise qPCR (as shown by the values in electronic supplementary material, table S1). To address the fact that we had two LTL measurements from each cohort, and in some analyses we needed to either compare the difference in SMDs between time-points (table 2, model 4), or combine SMDs across time-points (table 2, model 5), we used the methods outlined in Borenstein *et al.* [39, chapter 24] for computing effect sizes from complex data structures involving multiple time-points.

To facilitate interpretation of the summary SMDs derived from the meta-analyses we transformed these effects into base pairs. To make these calculations, estimates of the standard deviations of either LTL or LTL attrition rate were required, as appropriate. We estimated these standard deviations from the four cohorts measured with TRF (ADE, BHS, ERA and JLRCS), because these cohorts were the only ones for which LTL was originally measured in base pairs. We used weighted means of the four estimates of s.d. obtained from each cohort to account for the fact that the precision of an estimate of s.d. will increase with the square root of the sample size. The resulting estimates for the standard deviation of LTL and LTL attrition rate were 675.88 bp and 24.82 bp yr$^{-1}$ respectively.

# 3. Results

## 3.1. Characteristics of the dataset

We obtained data from 18 longitudinal cohorts spanning 10 countries and four continents (table 1 and electronic supplementary material, table S1). The combined dataset included data from 12 579 adults comprising 4678 current smokers and 7901 non-smokers. The mean age at baseline of the cohorts was 54.8 ± 15.4 years (mean ± s.d.; range: 26.0–80.2). Given that tobacco use typically begins by age 16 [41], smokers in the dataset are likely to have already been smoking for at least a decade at the time of the baseline telomere measurement. The mean follow-up interval was 8.6 ± 2.2 years (mean ± s.d.; range: 5.9–13.1).

Fourteen cohorts measured LTL using the quantitative polymerase chain reaction (qPCR) method and four used the terminal restriction fragment method (TRF; ADE, BHS, ERA and JLRCS; electronic supplementary material, table S1). The correlation between LTL at baseline and follow-up varied considerably among cohorts: Pearson correlation coefficients ($r$) ranged from 0 to 0.66 for qPCR measurements and from 0.92 to 0.97 for TRF measurements. These data suggest substantial variation in LTL measurement error among cohorts and greater measurement error in the subset of cohorts measured with qPCR.

## 3.2. Does leukocyte telomere length decrease with age?

To ascertain whether the telomere data were likely to be of sufficient quality to reveal effects of smoking on LTL dynamics, we first asked whether average LTL was shorter at follow-up than at baseline (when cohorts were a mean of 8.6 years younger). As expected, mean LTL was significantly shorter at follow-up, both in the meta-analysis and in all but one of the individual cohorts (table 2, model 1; electronic supplementary material, figure S2a). When we added the mean length of the follow-up interval to model 1 as a moderator, the slope of the meta-regression was in the expected direction, with longer follow-up intervals associated with larger declines in mean LTL between baseline and follow-up (electronic supplementary material, figure S2b). Although the meta-regression was not significant overall (parameter estimate and 95% CI: −0.02, [−0.09, 0.04], $p = 0.4218$), it was significant for

**Table 2.** Results from random-effects meta-analytic models on data from 18 cohorts.

| model | | summary SMD | | heterogeneity statistics | | | | | forest plot |
|---|---|---|---|---|---|---|---|---|---|
| | | estimate[a] [95% CI] | p-value | $Q_{17}$ | p-value | $\tau^2$ | $I^2$ (%) | $H^2$ | |
| 1 | difference in LTL between time-points | −0.49 [−0.62, −0.36] | <0.0001 | 333.73 | <0.0001 | 0.07 | 95.89 | 24.31 | electronic supplementary material, figure S2a |
| 2 | assn. between smoking and baseline LTL | −0.10 [−0.16, −0.05] | 0.0003 | 26.55 | 0.0650 | 0.01 | 44.70 | 1.81 | figure 2a |
| 3 | assn. between smoking and follow-up LTL | −0.15 [−0.29, −0.01] | 0.0312 | 95.13 | <0.0001 | 0.06 | 89.78 | 9.78 | figure 2b |
| 4 | difference in assn. between smoking and LTL between time-points | −0.05 [−0.16, 0.06] | 0.3884 | 73.45 | <0.0001 | 0.04 | 90.41 | 10.43 | figure 3a |
| 5 | combined assn. between smoking and LTL across time-points | −0.13 [−0.22, −0.03] | 0.0075 | 85.62 | <0.0001 | 0.03 | 83.96 | 6.24 | not shown |
| 6 | assn. between smoking and LTL attrition | −0.02 [−0.07, 0.04] | 0.5311 | 24.68 | 0.1021 | 0.00 | 36.48 | 1.57 | figure 4a |

[a]Negative parameter estimates for the summary standardized mean difference (SMD) correspond to: model 1—shorter LTL at follow-up; models 2, 3 and 5—shorter LTL in smokers; model 4—stronger assn. between smoking and LTL at follow-up; model 6—faster attrition in smokers. Significant associations are shown in italics.

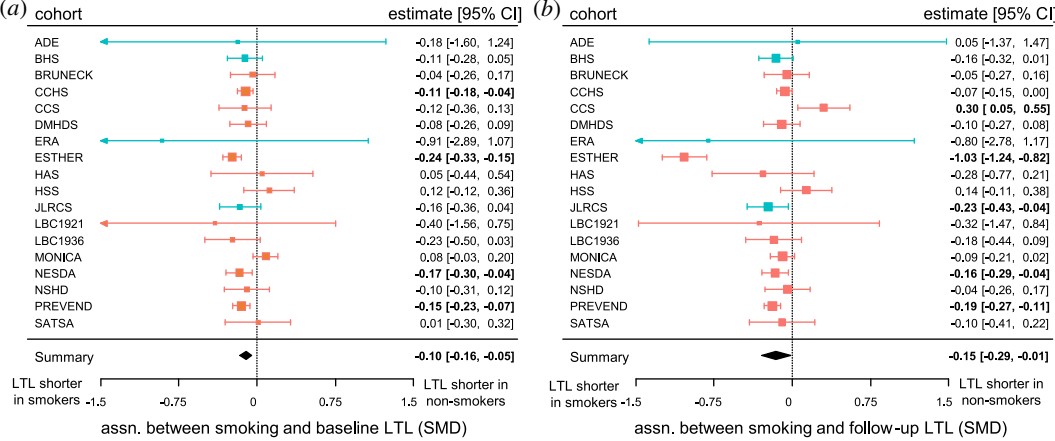

**Figure 2.** Smokers have shorter LTL than non-smokers at both baseline and follow-up. Forest plots showing the significant associations between smoking and LTL at (*a*) baseline (model 2), and (*b*) follow-up (model 3). The squares show the observed standardized mean difference (SMD) and the whiskers the 95% CI for each cohort; the area of each square is proportional to the weight given to that cohort in the meta-analysis. Cohorts measured with qPCR are shown in red and cohorts measured with TRF in blue. Significant differences are shown in bold. The black diamond shows the meta-analytic summary: the centre depicts the mean effect and the width the 95% CI.

the subset of cohorts measured with TRF (parameter estimate and 95% CI: $-0.03$, $[-0.05, <0.00]$, $p = 0.0186$). The estimated slope of the meta-regression was similar for both the whole dataset and the TRF subset, suggesting that although qPCR measurements are less precise, there is little evidence for bias.

Using the parameter estimate for the effect of age on LTL obtained from model 1 we calculated the estimated decrease in LTL between baseline and follow-up as 330.77 bp (95% CI: 241.55, 420.00). This equates to age-related attrition of 38.33 bp yr$^{-1}$ (95% CI: 27.99, 48.67), a value not significantly different from estimates of annual attrition obtained from longitudinal studies of LTL in cohorts not included in the current dataset (e.g. 40.2 bp yr$^{-1}$ [42] and 31.0 bp yr$^{-1}$ [43]). Thus, our data are of sufficient quality to show a robust effect of age on LTL of the expected magnitude over follow-up periods of as little as 5.9 years.

## 3.3. Do smokers have shorter leukocyte telomere length than non-smokers?

At baseline, smokers had shorter LTL than non-smokers in 17 cohorts and the difference was significant in four of these (figure 2*a*). At follow-up, smokers had shorter LTL in 15 cohorts and the difference was significant in four; non-smokers had significantly shorter LTL in one cohort (figure 2*b*). Meta-analyses showed that the association between smoking and LTL was significant overall at baseline and follow-up, with shorter LTL in smokers at both time-points (table 2, models 2 and 3 respectively). The causation and mixed hypotheses both predict that the difference in LTL between smokers and non-smokers should increase over time (figure 1). Three cohorts followed this pattern with a significantly stronger association between smoking and LTL at follow-up and one cohort showed the opposite pattern with a significantly stronger association at baseline, but for 14 of the cohorts, there was no difference in the association between smoking and LTL between baseline and follow-up (figure 3*a*). Meta-analysis of the difference in the association between smoking and LTL between time-points showed that this was not significantly different from zero overall (table 2, model 4). Thus, the pattern of cross-sectional findings within cohorts does not support the causation or mixed hypotheses (figure 1).

Since there was no evidence for a difference in association between time-points, we combined the data across time-points to obtain a single, more powerful, estimate of the cross-sectional association between smoking and LTL for each cohort. Combining across time-points yielded five cohorts with significantly shorter LTL in smokers, and meta-analysis confirmed significantly shorter LTL in smokers overall (table 2, model 5). In the current analysis, the observed association between smoking and LTL (from model 5) equates to smokers having telomeres 84.61 bp (95% CI: 22.62, 146.61) shorter than non-smokers.

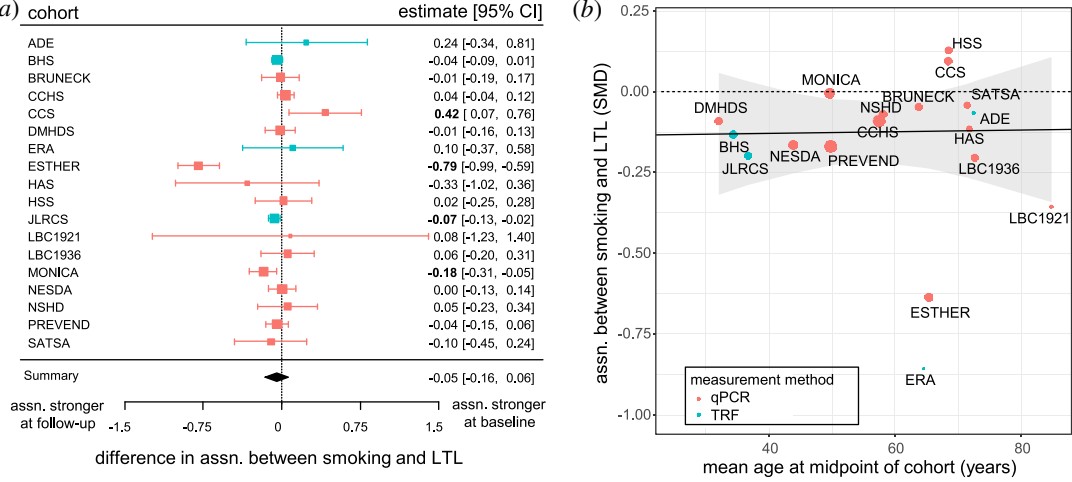

**Figure 3.** The difference in LTL between smokers and non-smokers does not increase with more years of smoking. (*a*) Forest plot showing the lack of difference in association between smoking and LTL between baseline and follow-up (model 4). For key see figure 2. (*b*) Scatterplot showing that the association between smoking and LTL does not change as the mean age of the cohort increases. Each point represents one cohort and the area of the point is proportional to the weight in model 5. The solid black line shows the non-significant estimate from a random-effects meta-regression model obtained by adding mean age of the cohort as a moderator to model 5. The ribbon shows 95% CI for the estimate. The dashed line indicates no effect of smoking on LTL.

There was significant heterogeneity among cohorts in the size of the association between smoking and LTL (table 2, model 5). Given that most smokers start smoking in their teenage years, the causal and mixed hypotheses both predict that the difference in LTL between smokers and non-smokers should increase as the mean age of the cohort increases. Our dataset allows a powerful test of this prediction, because mean age at baseline varies from 26 (in DMHDS) to over 80 years (in LBC1921; a far greater age range than the effect of follow-up interval analysed in model 4). Therefore, we added mean age at the midpoint between baseline and follow-up as a moderator to model 5. The slope of the resulting meta-regression was not significantly different from zero (parameter estimate and 95% CI: $<0.00$ [$-0.01$, 0.01], $p = 0.9450$; figure 3*b*). Thus, there is no evidence that the difference in LTL between smokers and non-smokers becomes larger in older participants who are likely to have smoked for longer. In conclusion, neither the within-cohort nor the among-cohort effects of age show the patterns in LTL predicted by the causation and mixed hypotheses (figure 1).

## 3.4. Do smokers have faster telomere attrition than non-smokers?

We used the longitudinal LTL attrition rates of participants to ask whether LTL attrition was faster in smokers than in non-smokers. Two cohorts (JLRCS and MONICA) had significantly faster LTL attrition in smokers, while one cohort (CCS) had significantly faster LTL attrition in non-smokers (figure 4*a*). Meta-analysis showed no significant association between smoking and the rate of LTL attrition overall, and no significant heterogeneity in the size of the association between cohorts (table 2, model 6). The parameter estimate from model 6 equates to a difference in LTL attrition rate between smokers and non-smokers of $-0.51$ bp yr$^{-1}$ (95% CI: $-2.09$, 1.08). This difference in rate of attrition due to smoking is a negligible proportion (1.32%) of the 38.32 bp loss per year that we estimated for ageing.

To establish the robustness of the lack of an association between smoking and LTL attrition, we conducted three further analyses of the rate of LTL attrition. First, to test whether any single cohort was having a substantial influence on the association between smoking and LTL attrition, we explored the effect of excluding each cohort from model 6 in turn. In all cases, the resulting parameter estimate for the association between smoking and LTL attrition was not significantly different from zero (electronic supplementary material, table S2). The inclusion of just one cohort (MONICA) is responsible for the parameter estimate being negative; when this cohort is excluded, the parameter estimate for the association between smoking and telomere attrition was effectively zero (parameter estimate and 95% CI: 0.00 [$-0.05$, 0.05], $p = 0.9947$).

Second, to test whether the lack of an association between smoking and LTL attrition is explained by the inclusion of cohorts with high LTL measurement error in the dataset, we re-ran model 6 weighting

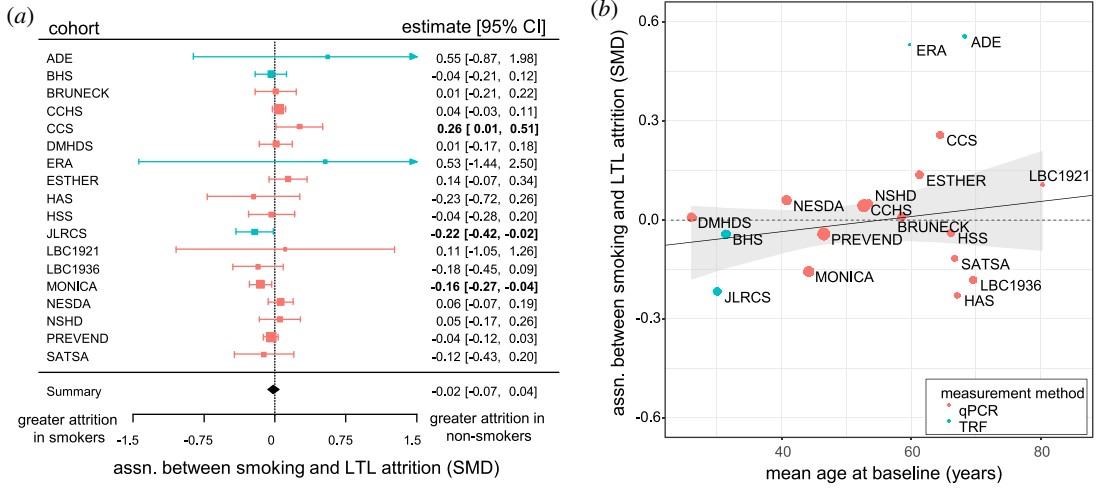

**Figure 4.** The rate of LTL attrition is virtually identical in smokers and non-smokers and this absence of a difference in attrition does not change over 54 years of smoking. (*a*) Forest plot showing the lack of an association between smoking and LTL attrition rate measured longitudinally within participants (model 6). For key see figure 2. (*b*) Scatterplot showing the lack of association between the effect of smoking on LTL attrition and mean age at baseline. The solid black line shows the non-significant estimate from a random-effects meta-regression model obtained by adding mean age at baseline as a moderator to model 6. The dashed line indicates no association between smoking and LTL attrition. For key see figure 3*b*.

the contribution of each cohort by the correlation between baseline and follow-up LTL measurements (*r*). The resulting parameter estimate for the association between smoking and telomere attrition was now slightly positive, but still not significantly different from zero (parameter estimate and 95% CI: 0.08 [−0.20, 0.36], *p* = 0.5694).

Third, we explored whether there was any evidence for bias in our selection of cohorts. We started with a systematic literature search, but due to the lack of suitable data in published articles, the final dataset assembled is an opportunity sample. For half of the cohorts included we had some idea of the effects of smoking expected due to information available in published articles [17], but for the remaining half we were blind to the expected findings at the time of requesting the data (table 1). Re-running model 6 on just the latter nine cohorts produced no change in the conclusion that smoking does not affect LTL attrition rate (parameter estimate and 95% CI: −0.04 [−0.17, 0.09], *p* = 0.5797).

## 3.5. Can the small difference in attrition explain the cross-sectional difference in leukocyte telomere length?

Although the difference in attrition between smokers and non-smokers was negligible and not significantly different from zero, attrition was still slightly faster in smokers (i.e. a negative parameter estimate) when we included all cohorts in the meta-analysis and weighted them in the conventional fashion (inverse-variance). Using the estimates of the difference in LTL from model 5 and the difference in attrition from model 6, we asked how many years of smoking would be necessary to generate the cross-sectional difference in LTL between smokers and non-smokers. The parameter estimates from models 5 and 6 in table 2 suggest that LTL is 84.61 bp shorter in smokers compared to non-smokers and that LTL attrition rate 0.51 bp yr$^{-1}$ faster in smokers than non-smokers. Thus, assuming a linear effect of smoking on attrition, 167.43 years of smoking would be required to generate the observed cross-sectional difference in LTL between smokers and non-smokers. Note that when we weighted cohorts by their measurement error in model 6 (see above), LTL attrition was estimated as 2.29 bp yr$^{-1}$ faster in non-smokers than smokers. Under this scenario, no number of years of smoking can generate the observed baseline difference in LTL.

## 3.6. Is the effect of smoking on the rate of leukocyte telomere length attrition nonlinear?

Thus far, our tests of the causation hypothesis have assumed a sustained effect of smoking on the rate of LTL attrition that continues unabated in smokers (as illustrated in figure 1). To test whether an effect of

smoking on the rate of LTL attrition is present in newer smokers and diminishes or reverses over time with continued smoking, we added age at baseline as a moderator to model 6. The slope of the resulting meta-regression was not significantly different from zero (parameter estimate and 95% CI: <0.00 [−0.00, 0.01], $p = 0.3019$; figure 4b). Thus, there is no evidence for an association between smoking and LTL attrition in younger people (who have more recently started smoking) that subsequently diminishes over time.

# 4. Discussion

Using meta-analytic methods to compare LTL dynamics over a mean of 8.6 years of follow-up in 4678 current smokers and 7901 non-smokers, we found no evidence for significantly faster telomere attrition in smokers. Our analyses confirmed that LTL shortened with increasing age and that smokers had shorter LTL than non-smokers at all measured time-points. However, there was no evidence that this cross-sectional difference increased with age, as would be expected if smoking causes LTL attrition. There was also no significant difference in the rate of LTL attrition measured within smokers and non-smokers. Moreover, the negligibly greater rate of attrition observed in smokers was totally insufficient to produce the cross-sectional difference between smokers and non-smokers within a human lifetime. Taken together, these findings provide no support for the hypothesis that smoking causes a sustained increase in the rate of LTL attrition.

The above conclusion is based on assuming a linear effect of smoking on telomere attrition over time (figure 1). There are strong mechanistic reasons for assuming a linear model. Current smoking causes a chronic elevation in the levels of oxidative stress and inflammation, both of which are implicated in accelerated telomere attrition in a dose-dependent manner. However, some authors have attempted to explain the lack of an effect of smoking on telomere attrition in previous studies by assuming that smoking has an initial accelerating effect on attrition that rapidly reverses with continued smoking [20–22]. The evidence claimed for this nonlinear effect is the strong positive correlation observed in all longitudinal studies between baseline telomere length and telomere attrition: individuals starting with longer telomeres have faster attrition than those starting with shorter telomeres. On the basis of this observation, Farzaneh-Far et al. [22] have argued that any rapid initial shortening of telomeres caused by smoking is subsequently offset by a length-related decrease in attrition caused by this 'homeostatic' process. However, it has now been recognized that a more parsimonious explanation for the observed correlation between baseline LTL and attrition rate is regression to the mean resulting from measurement error [29]. Thus, the apparent 'homeostatic' mechanism proposed to account for nonlinear attrition rates in smokers is largely a statistical artefact. Importantly, our analysis of the current dataset provided no evidence that the association between smoking and LTL attrition changed over time, as would be expected if an accelerating effect of smoking was present early on and subsequently diminished or reversed. It is a limitation of our dataset that our youngest cohorts were already in their 20s, meaning that the majority of smokers are likely to have been smoking for at least a decade. It is therefore possible that we could have missed an effect of smoking on LTL attrition that occurred prior to the baseline telomere measurements. To decisively rule out a highly nonlinear effect of smoking that is completely restricted to the decade immediately after starting smoking, we need to know whether telomere length differences in childhood precede the initiation of smoking [17]. Cohorts are currently becoming available in which it will be possible to test this prediction.

Is the lack of an effect of smoking on telomere attrition a limitation of low power? Some authors have argued that longitudinal studies have low power for detecting effects of smoking on attrition due to their small sample sizes [20,21,26]. While it is often the case that sample sizes are larger in cross-sectional than longitudinal studies, this is not true of the 18 cohorts included in the current analysis, where the same individuals were studied both cross-sectionally and longitudinally. Furthermore, longitudinal studies eliminate the substantial between-individual variation in LTL by measuring within-individual changes in LTL and are therefore much more powerful than cross-sectional studies of the same sample size for detecting effects of smoking on telomere dynamics [44]. Importantly, our meta-analysis of 18 longitudinal datasets shows no significant heterogeneity among cohorts in the effect of smoking on telomere attrition rates. This suggests first, that it is valid to compute a summary estimate for the difference in attrition between smokers and non-smokers, and second, that the precision of the meta-analysis exceeds that of its constituent cohorts [39]. The resulting negligible difference in attrition between smokers and non-smokers of $−0.51$ bp $yr^{-1}$ is therefore the most powerful estimate yet of the association between smoking and telomere attrition.

What is the likely impact of telomere measurement error on our findings? The variation in the correlation between baseline and follow-up LTL measurements suggests substantial variation in measurement error among cohorts [27]. While this error should not cause bias in our estimates of the association between smoking and LTL attrition, it will affect the precision of these estimates [30]. We found no evidence that weighting cohorts according to the correlation between baseline and follow-up measurements caused a significant change in our estimate of the association between smoking and LTL attrition. Indeed, the weighted meta-analysis actually yielded a marginally lower rate of LTL attrition in smokers compared to non-smokers. This result strengthens the evidence against the causation hypothesis, because it implies that no amount of smoking could yield the observed cross-sectional difference in LTL.

Is it possible that some kind of bias is masking a true effect of smoking on telomere attrition? Restricting our dataset to participants that survived to follow-up undoubtedly introduces selection based on mortality (e.g. [26]). Since both smoking and short LTL have been argued to cause earlier mortality, mortality is a collider variable in our analyses. Selection based on the value of a collider is usually discussed in the context of producing spurious associations between independent variables, so-called 'collider bias' (e.g. [45]), but collider bias can also potentially mask true associations. For example, by selecting against individuals who die between baseline and follow-up, longitudinal studies could underestimate the effects of smoking on telomere attrition, because they retain only smokers who are resistant to the damaging effects of tobacco smoke. However, this argument fails to explain the substantial difference in baseline LTL between smokers and non-smokers that we observed, even in the studies where the age of participants at baseline was quite advanced. If selection bias is present, it should affect cross-sectional associations as well as measures of attrition, yet we found no change in the difference in LTL between smokers and non-smokers with increasing cohort age (figure 3b). Furthermore, selection based on mortality should be negligible in cohorts in their 20s or 30s, but substantial for cohorts over 60, yet we found no effect of baseline age on the size of the association between smoking and LTL attrition, despite an age range of over 54 years. Taken together, the above findings argue against selection bias masking a true association between smoking and LTL attrition.

We deliberately elected to use raw attrition measures in our analyses as opposed to effect sizes derived from multiple regression models controlling for known sources of variation in telomere attrition rates (such as baseline telomere length). Our rationale for this choice came from recent work showing that controlling for baseline telomere length in multiple regression models of telomere attrition biases the effects of any predictor variables that also correlate with telomere length at baseline (typically age, sex and smoking status; [30]). This latter finding suggests that the published effects of variables such as age, sex and smoking status on telomere attrition are likely to exaggerate the true effect sizes of these variables, raising the probability of type I errors above 5% (see also [46]). This could explain why some of our individual cohorts report significant effects of smoking on LTL attrition [18,19].

A corollary of our decision to use raw LTL and attrition measures in our analyses is that we did not control for potential confounds including age and sex. However, the majority of the nine published studies that we have been able to find reporting effects of smoking on telomere attrition use multiple regression models that control for age and sex [18–26]. Considered together, the results of these nine studies support the general conclusions of the current paper: there is strong evidence for an effect of smoking on LTL (six out of eight studies that tested for a difference report that LTL is significantly shorter in smokers), but there is much less evidence for an effect of smoking on LTL attrition (only two out of nine studies report that LTL attrition is significantly faster in smokers) (see [17, table 3]). Furthermore, it is reassuring that our meta-analysis of the cross-sectional effect of smoking on LTL produces a summary effect size for smoking ($-0.13$) that is similar to that reported in another meta-analysis based on published effect sizes derived from cross-sectional studies that control for potential confounds such as age and sex ($-0.11$ in [1]). Thus, the conclusions drawn from analyses that do and do not control for age and sex appear very similar and there is no evidence to suggest that controlling for age and sex would alter the conclusions of the current paper.

In the absence of any evidence supporting the hypothesis that smoking causes a sustained increase in the rate of LTL attrition, it is worth considering the alternative hypothesis that selective adoption is occurring. Selective adoption predicts that a difference in LTL between future smokers and non-smokers should exist prior to the start of smoking. Two alternative causal pathways could underlie selective adoption [17]. First, it is possible that telomere shortening could directly cause changes in behaviour. There is emerging evidence that telomere shortening causes changes in regulation of more

than 140 genes [47,48], making this idea theoretically possible (although none of the genes identified thus far as regulated by telomere shortening is obviously linked to behaviour, let alone smoking). A second causal pathway yielding selective adoption is that exposure to a third variable both shortens LTL and makes subsequent adoption of smoking more likely. One possibility, that still attributes a causal role to smoke exposure, is that parental smoking both causes early-life LTL attrition and increases a child's probability of starting smoking. A recent study found that telomere loss between birth and young adulthood was positively associated with distance to a major road at the residential address occupied at birth [49], suggesting air pollution as a possible cause of childhood telomere attrition. Thus, it is possible that passive smoking in early life could cause telomere attrition. However, it is not necessary to attribute any causal role to smoke exposure to explain the data in the current paper. We suggest that a plausible third variable supported by substantial existing data is exposure to early-life adversity. Developmental telomere attrition is accelerated by exposure to early-life adversity of various types including family disruption and physical and emotional abuse [50–53]. Furthermore, these same sources of early-life adversity are also associated with a greater probability of starting smoking, smoking more and being less likely to quit [54–56]. Thus, although childhood LTL has not thus far been examined as a predictor of adult smoking behaviour, there is strong indirect evidence to expect associations to exist. It is worth noting that the available data lead us to predict not only an association between childhood LTL and the presence of adult smoking, but also between childhood LTL and the amount smoked. An association between LTL and amount smoked is often regarded as strong evidence for the causation hypothesis [1]. However, it is now clear that such evidence is equally compatible with selective adoption.

In conclusion, we find no evidence that smoking accelerates the rate of leucocyte telomere attrition in adults. Our findings should prompt more critical appraisal of data underlying the claim that smoking is the most important, 'broad range' ageing accelerator [5,6]. Where these data come from cross-sectional studies, and *in vivo* experimental studies are lacking, we suggest that selective adoption should be considered as an alternative explanation for associations between smoking and biomarkers of ageing such as telomere length.

Our findings have consequences for how measures of telomere length are used in human epidemiology and behavioural ecology. Under the currently prevailing view that certain types of behaviour cause accelerated telomere attrition, measures of telomere length can be used to identify those behaviours that are most harmful and those that are protective [57]. Changes in telomere dynamics could also potentially be used to monitor the somatic consequences of behaviour change (e.g. the positive effects of quitting smoking). However, if we are correct, and selective adoption turns out to be an explanation for observed associations with telomere length, then we need to reinterpret shorter telomeres as a relatively static biomarker as opposed to as a dynamic consequence of current adult behaviour.

As a final note, although we found no evidence that smoking accelerates the rate of leucocyte telomere attrition, our results do not preclude the many other well-established negative effects of smoking on human health and longevity. We chose to focus on smoking in the current paper simply because there are more data available on the associations between smoking and telomere length than for any other behaviour [17]. Our intention was to question prevalent assumptions in the telomere dynamics literature concerning the mechanisms underlying associations between behaviour and telomere length, rather than to question the damaging effects of smoking.

Ethics. No additional ethical permission was required for this study because the analysis was based on secondary analysis of existing anonymized data.

Data accessibility. An R script that generates the figures and analyses presented in this paper and the required summary dataset (also presented in electronic supplementary material, table S1) are publicly available at: doi:10.5281/zenodo. 1240964. Individual participant-level data for CCS, HAS, LBC1921, LBC1936 and NSHD are available on request to bona fide researchers. For NSHD see http://www.nshd.mrc.ac.uk/data; doi:10.5522/NSHD/Q101 and doi:10. 5522/NSHD/Q102).

Authors' contributions. Conceived the study, analysed the data and wrote the paper: M.B. and D.N.; assisted in identification of cohorts: C.M.M.-R. and G.V.P.; provided summary statistics: F.K. and P.W. (BRUNECK), S.E.B. and B.G.N. (CCHS), W.M.T. (DMHDS), C.M. and M.W. (HSS), A.A. and J.D.K. (JLRCS), L.B. (MONICA), D.R. and B.W.J.H.P. (NESDA), P.v.d.H. and M.A.S. (PREVEND), S.H. and Y.Z. (SATSA); provided raw data: A.B. and C.L. (ADE, ERA), Y.B.-S. (CCS), C.C. and H.S. (HAS), R.C. and D.K. (NSHD), I.J.D., S.E.H. and J.M.S. (LBC1921 and LBC1936); ran original LTL measurements: C.M.M.-R. and T.v.Z. (CCS, HAS, LBC1921, LBC1936, NSHD); edited and approved the final manuscript: all authors.

Competing interests. The authors declare no competing interests.

Funding. Cohorts acknowledge the following sources of funding: LBC1921, CCS, HAS and NSHD—New Dynamics of Ageing via the HALCyon cross-cohort collaborative programme (RES-353-25-0001); LBC1921—UK Biotechnology and Biological Sciences Research Council (BBSRC), The Royal Society, and The Chief Scientist Office of the Scottish Government, Lifelong Health and Wellbeing Initiative (MR/K026992/1); LBC1936—Age UK (Disconnected Mind Project); NSHD—UK Medical Research Council; JLRCS—US-Israel Binational Science Foundation, the Israel Science Foundation and the National Institutes of Health (AG030678 and AG201320); DMHDS—US National Institute on Aging (AG032282) and the UK Medical Research Council (MR/K00381X and MR/P005918); NESDA—The Netherlands Organisation for Health Research and Development (10-000-1002), VU University Medical Center, GGZ inGeest, Leiden University Medical Center, Leiden University, GGZ Rivierduinen, University Medical Center Groningen, University of Groningen, Lentis, GGZ Friesland, GGZ Drenthe, Rob Giel Onderzoekscentrum. The authors additionally acknowledge the following sources of funding: M.B.—National Centre for the Replacement Refinement and Reduction of Animals in Research (NC/ K000802/1); D.N. and G.V.P.—European Research Council (AdG 666669); A.A.—National Institutes of Health (R01HL116446, R01HD071180, R01HL13840); S.H. and Y.Z.—Karolinska Institutet Delfinansiering, the Swedish Research Council (2015-03255), the Loo & Hans Osterman Foundation, the Foundation for Geriatric Diseases, the Magnus Bergwall Foundation, the Erik Rönnberg award for aging studies and the Strategic Research Program in Epidemiology at Karolinska Institutet; C.C. and H.S.—UK Medical Research Council and University of Southampton; T.v.Z.—UK Medical Research Council (G0601333); I.J.D., J.M.S. and S.E.H.—Centre for Cognitive Ageing and Cognitive Epidemiology, which is funded by the Medical Research Council and the Biotechnology and Biological Sciences Research Council (MR/K026992/1).

Acknowledgements. We are grateful to all of the people who took part in the study, either as participants, or as part of the research teams responsible for the 18 cohorts analysed. We thank Jimmy Zeng who computed summary statistics for DMHDS. We also thank Maya B. Mathur and Idan Shalev who provided thoughtful reviewers' comments that substantially improved the paper.

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
