## [Reviewer comments · Royal Society Open Science]

Review History

RSOS-181763.R0 (Original submission)

Review form: Reviewer 1 (Maya Mathur)

Is the manuscript scientifically sound in its present form?

Yes

Are the interpretations and conclusions justified by the results?

Yes

Is the language acceptable?

Yes

Is it clear how to access all supporting data?

Yes

Do you have any ethical concerns with this paper?

No

Have you any concerns about statistical analyses in this paper?

No

Recommendation?

Accept as is

Comments to the Author(s)**OVERALL COMMENTS**

This exemplary meta-analysis was a pleasure to read. The design and statistical methods evidence careful attention; I was pleased with the use of broad search terms likely to capture all relevant published literature, the inclusion of only longitudinal studies with adequate follow-up, the use of directly comparable statistical estimates from each study (which often required contacting the original authors or re-analyzing published data), and the sensible, thorough statistical reporting. The Discussion and sensitivity analyses show careful consideration of possible alternative explanations. I furthermore confirm that the authors have publicly provided well-documented, clear code and data to reproduce their results.

I therefore believe that the primary finding that smoking is not longitudinally associated with telomere attrition is an excellent example of a persuasively null result. I would be happy to see the paper accepted in its present form, as I have only the following minor suggestions.

MINOR COMMENTS

- 1.) Methods: Were there any differences in the analyses conducted in studies in which the authors had access to individual participant data (IPD) vs. only summary statistics? For example, the penultimate paragraph of "Methods" describes subject-level exclusion criteria. Were these same criteria applied when only summary estimates were available?
- 2.) Methods: Given that IPD were available for 9 cohorts, would it have been possible to conduct counterpart subject-level analyses on this subset of the data? Was this infeasible due to different measurement scales for the LTL variables?
- 3.) Methods, final paragraph: I initially wasn't sure what "pooled LTL" referred to, though I eventually inferred that it meant "pooled across time points".
- 4.) Results, final paragraph, and page 9: I am not sure that variation in the correlation between baseline and F/U LTL is necessarily evidence for measurement error. Couldn't it simply be that some cohorts have more inherent variation in LTL attrition than others, for example due to demographic or behavioral characteristics?
- 5.) Page 12, "While there is no data relating specifically to genes linked to behavior": Do you mean telomeres rather than "genes" broadly speaking? Of course genes are linked to behavior.

6.) Very last paragraph: Do you perhaps mean “selective adoption” instead of “selective causation”?

Signed,
Maya B. Mathur
PhD Biostatistics
Department of Epidemiology
Harvard University

Review form: Reviewer 2 (Idan Shalev)

Is the manuscript scientifically sound in its present form?

Yes

Are the interpretations and conclusions justified by the results?

No

Is the language acceptable?

Yes

Is it clear how to access all supporting data?

Yes

Do you have any ethical concerns with this paper?

No

Have you any concerns about statistical analyses in this paper?

No

Recommendation?

Major revision is needed (please make suggestions in comments)

Comments to the Author(s)

The authors present a meta-analysis of 18 cohort studies with longitudinal data on telomere length and smoking status. Prevailing thinking assumes a causal link exists between smoking and accelerated telomere shortening. However, this claim is based largely on cross-sectional data. The current manuscript provocatively challenges this view, presenting a fresh and alternative hypothesis of ‘selective adoption’ that assumes individuals with shorter telomere length are more likely to start smoking, and that some other factors both shorten telomeres and predispose individuals toward smoking behavior. The paper is well-written and theoretically sound. However, the manuscript can be improved prior to publication by addressing several conceptual, inferential and methodological issues.

1. Inferential: The current work presents 3 key findings: a) LTL in smokers is shorter than LTL in non-smokers at baseline and follow up, b) LTL decreases over the span of time between baseline and follow up, and c) the rate of LTL attrition between baseline and follow-up is not significantly different in smokers relative to non-smokers. These findings are explained in the context of the ‘causal’ and ‘selective adoption’ hypotheses for the effect of smoking on telomere length. However, the results of this meta-analysis neither prove nor disprove the causal or selective adoption hypotheses since no LTL measurements exist prior to the onset of smoking. For the most part, the authors are careful in their wording of what conclusions can be derived from their

work. However, there are a few places where uses of strong language imply that the data completely support the selective adoption hypothesis. Overall, the authors are recommended to use a more suggestive language by removing any (implied) causal inferences.

For example, page 12: “Our findings do not support the hypothesis that smoking is the sole cause of differences in LTL between smokers and non-smokers. Accordingly, we conclude that selective adoption of smoking by individuals with short telomeres is likely to be occurring”. The data show shorter LTL in smokers versus non-smokers across two time points, and thus, provides no evidence for what may be causing the difference. It could be smoking if the effect of smoking had completely manifested before baseline LTL measurements. In fact, smoking may very well be one of the causes since it is the only variable different in these individuals at baseline other than their LTL (since nothing else was controlled for). Furthermore, beginning the next sentence with “accordingly” implies an “if not A then B” logic to the relationship. In other words, the passage implies that if LTL is not wholly shortened by smoking then selective adoption must be happening.

2. Conceptual: While the ‘selective adoption’ hypothesis is interesting, and plausible, perhaps the more parsimonious explanation for the ‘selective adoption’ could simply be a confounding variable? Whether it is, for example, prior exposure to early-life adversity, low SES, health status, alcohol use, or just education, specifically. For example, when controlling for SES, the effect of smoking is usually reduced. Further, the potential relationship between LTL and smoking through early-life adversity could potentially be tested using other datasets such as the UK Environmental Risk cohort with repeated telomere length measurements prior to smoking at ages 5, 10 and 18 years. This study, which includes variables such as exposure to violence and telomere length prior to smoking initiation and smoking status at age 18 years, can be used to test the authors’ ‘selective adoption’ hypothesis.

3. Methodological: The argument for the exclusion of the approximately 1,200 people that had ambiguous smoking status would be strengthened with descriptive statistics for this group (e.g. age, smoking status change, and other variables common between the studies included in this meta-analysis). If significant differences exist in these individuals, a bias may be introduced. For instance, individuals around the age of the full sample mean (~55 years) may be starting to see the detrimental health impacts of long-term smoking and be enticed to quit during the sampling period. The individuals most impacted by smoking may be the ones most likely to quit, thus introducing a potential bias by their exclusion. Individuals that can smoke into old-age and stay reasonably healthy while doing so (due to protective factors, genetics, etc.) might be overrepresented in this sample. The cross-sectional relationship between smoking and LTL is typically very small and could easily be overwhelmed by such biases.

4. Methodological: A key feature of Hill’s (1965) seminal work on causality in epidemiological research is biological gradient, wherein increased exposure results in the increased likelihood and severity of the outcome. The authors might benefit from the inclusion of information about dose response (i.e., pack-year of smoking, as in Astuti et al. 2017) into the manuscript and how it might impact the results.

5. Methodological: Three cohorts, ADE (2 vs. 42), ERA (1 vs. 86), & LBC1921 (3 vs. 76), had overwhelmingly unbalanced distributions of smokers and nonsmokers. These studies are severely underpowered to make claims about differences in LTL between smokers and non-smokers. This limitation is not discussed, despite its impacts on several key aspects of the study highlighted below. The manuscript could be improved by adding supplementary analyses excluding these three studies.

a. Two of these studies (ADE & ERA) comprise half of the studies using the TRF method, estimates of which were used to convert SMDs to units of base pairs. While this conversion may

be defensible as described based on strengths of the measurement method used (e.g. TRF relative to qPCR), taking standard deviation estimates of LTL and LTL attrition from cohorts containing only one to three smokers and applying these estimates to SMDs from other studies with over 1,000 smokers seems inappropriate.

b. The same two cohorts (ADE & ERA) also had high correlation between baseline and follow-up measurements. Thus, the authors' decision to rerun model 6, "weighting the contribution of each cohort by the correlation between baseline and follow-up LTL measurements" (page 9), grossly overvalues the contribution of these studies toward the overall effects observed across all 18 studies.

6. Analysis: Was the impact of chronological age on LTL controlled for in the analyses investigating differences in LTL between smokers and non-smokers at baseline and follow-up (i.e. Models 2 and 3)? The only models in which chronological age are explicitly stated to be included are Models 1 and 5.

7. Analysis: Age-adjusted LTL measures should similarly be used to recalculate LTL attrition rates. As currently described, LTL attrition rates were calculated without controlling for any factors at baseline, including baseline LTL. The latter was excluded because, "the observed correlation between baseline LTL and attrition rate is regression to the mean resulting largely from measurement error" (pages 10-11). This statistical rationale is tenable. However, other control variables exist as the authors themselves acknowledged on page 12: "controlling for baseline telomere length in multiple regression models of telomere attrition biases the effects of other predictor variables that correlate with telomere length at baseline like age, sex, and smoking status." However, the authors do not go on to account for these variables in their models. In other words, I wonder whether the effects of these other variables are exaggerated because they are not considered at all? I suggest the authors reconsider their adjustment for age and sex at baseline and follow-up in their analyses of LTL attrition rates.

Miscellaneous comments:

- In describing their literature search, the authors note the use of snowballing. For snowballing to be done adequately, the set of papers from which the authors attempted to move forward and backward through needed to be carefully vetted. Including the PRISMA flow diagram (how many papers screened etc.) will improve the transparency of the data presentation.

-There is no limitation section included in the discussion. Some of the topics highlighted above can be included in this section (i.e., selective attrition of smokers, potential sex differences, SES, dose-response etc.) Doing so would help transform the otherwise (implied) causal language used to support the selective adoption hypothesis into more appropriate tentative conclusions.

Idan Shalev
Penn State University

Decision letter (RSOS-181763.R0)

25-Jan-2019

Dear Dr Bateson:

Manuscript ID RSOS-181763 entitled "Smoking does not accelerate leukocyte telomere attrition: a meta-analysis of 18 longitudinal cohorts" which you submitted to Royal Society Open Science, has been reviewed. The comments from reviewers are included at the bottom of this letter.

In view of the criticisms of the reviewers, the manuscript has been rejected in its current form. However, a new manuscript may be submitted which takes into consideration these comments.

Please note that resubmitting your manuscript does not guarantee eventual acceptance, and that your resubmission will be subject to peer review before a decision is made.

Your resubmitted manuscript should be submitted by 25-Jul-2019. If you are unable to submit by this date please contact the Editorial Office.

on behalf of Dr John Dalton (Associate Editor) and Kevin Padian (Subject Editor)
openscience@royalsociety.org

Editor Comments:

We have somewhat divergent reviews on this MS, both of them thoughtful. Dr. Shalev in particular brings up some important issues for consideration that may require re-analysis and discussion among the (many) authors. Our timeframe for a simple "revise" decision is short and so mainly for this reason I log a "reject/resubmi" recommendation. However it will be important to address directly all of Dr. Shalev's comments as well as those of the other reviewer. Thanks for submitting.

Reviewers' Comments to Author:

Reviewer: 1

Comments to the Author(s)

OVERALL COMMENTS

This exemplary meta-analysis was a pleasure to read. The design and statistical methods evidence careful attention; I was pleased with the use of broad search terms likely to capture all relevant published literature, the inclusion of only longitudinal studies with adequate follow-up, the use of directly comparable statistical estimates from each study (which often required contacting the original authors or re-analyzing published data), and the sensible, thorough statistical reporting. The Discussion and sensitivity analyses show careful consideration of possible alternative explanations. I furthermore confirm that the authors have publicly provided well-documented, clear code and data to reproduce their results.

I therefore believe that the primary finding that smoking is not longitudinally associated with telomere attrition is an excellent example of a persuasively null result. I would be happy to see the paper accepted in its present form, as I have only the following minor suggestions.

MINOR COMMENTS

- 1.) Methods: Were there any differences in the analyses conducted in studies in which the authors had access to individual participant data (IPD) vs. only summary statistics? For example, the penultimate paragraph of “Methods” describes subject-level exclusion criteria. Were these same criteria applied when only summary estimates were available?
- 2.) Methods: Given that IPD were available for 9 cohorts, would it have been possible to conduct counterpart subject-level analyses on this subset of the data? Was this infeasible due to different measurement scales for the LTL variables?
- 3.) Methods, final paragraph: I initially wasn’t sure what “pooled LTL” referred to, though I eventually inferred that it meant “pooled across time points”.
- 4.) Results, final paragraph, and page 9: I am not sure that variation in the correlation between baseline and F/U LTL is necessarily evidence for measurement error. Couldn’t it simply be that some cohorts have more inherent variation in LTL attrition than others, for example due to demographic or behavioral characteristics?
- 5.) Page 12, “While there is no data relating specifically to genes linked to behavior”: Do you mean telomeres rather than “genes” broadly speaking? Of course genes are linked to behavior.
- 6.) Very last paragraph: Do you perhaps mean “selective adoption” instead of “selective causation”?

Signed,
Maya B. Mathur
PhD Biostatistics
Department of Epidemiology
Harvard University

Reviewer: 2

Comments to the Author(s)

The authors present a meta-analysis of 18 cohort studies with longitudinal data on telomere length and smoking status. Prevailing thinking assumes a causal link exists between smoking and accelerated telomere shortening. However, this claim is based largely on cross-sectional data. The current manuscript provocatively challenges this view, presenting a fresh and alternative hypothesis of ‘selective adoption’ that assumes individuals with shorter telomere length are more likely to start smoking, and that some other factors both shorten telomeres and predispose individuals toward smoking behavior. The paper is well-written and theoretically sound. However, the manuscript can be improved prior to publication by addressing several conceptual, inferential and methodological issues.

1. Inferential: The current work presents 3 key findings: a) LTL in smokers is shorter than LTL in non-smokers at baseline and follow up, b) LTL decreases over the span of time between baseline and follow up, and c) the rate of LTL attrition between baseline and follow-up is not significantly different in smokers relative to non-smokers. These findings are explained in the context of the ‘causal’ and ‘selective adoption’ hypotheses for the effect of smoking on telomere length. However, the results of this meta-analysis neither prove nor disprove the causal or selective adoption hypotheses since no LTL measurements exist prior to the onset of smoking. For the most part, the authors are careful in their wording of what conclusions can be derived from their work. However, there are a few places where uses of strong language imply that the data completely support the selective adoption hypothesis. Overall, the authors are recommended to use a more suggestive language by removing any (implied) causal inferences.

For example, page 12: “Our findings do not support the hypothesis that smoking is the sole cause of differences in LTL between smokers and non-smokers. Accordingly, we conclude that selective adoption of smoking by individuals with short telomeres is likely to be occurring”. The data show shorter LTL in smokers versus non-smokers across two time points, and thus, provides no evidence for what may be causing the difference. It could be smoking if the effect of smoking had completely manifested before baseline LTL measurements. In fact, smoking may very well be one of the causes since it is the only variable different in these individuals at baseline other than their LTL (since nothing else was controlled for). Furthermore, beginning the next sentence with “accordingly” implies an “if not A then B” logic to the relationship. In other words, the passage implies that if LTL is not wholly shortened by smoking then selective adoption must be happening.

2. Conceptual: While the ‘selective adoption’ hypothesis is interesting, and plausible, perhaps the more parsimonious explanation for the ‘selective adoption’ could simply be a confounding variable? Whether it is, for example, prior exposure to early-life adversity, low SES, health status, alcohol use, or just education, specifically. For example, when controlling for SES, the effect of smoking is usually reduced. Further, the potential relationship between LTL and smoking through early-life adversity could potentially be tested using other datasets such as the UK Environmental Risk cohort with repeated telomere length measurements prior to smoking at ages 5, 10 and 18 years. This study, which includes variables such as exposure to violence and telomere length prior to smoking initiation and smoking status at age 18 years, can be used to test the authors’ ‘selective adoption’ hypothesis.

3. Methodological: The argument for the exclusion of the approximately 1,200 people that had ambiguous smoking status would be strengthened with descriptive statistics for this group (e.g. age, smoking status change, and other variables common between the studies included in this meta-analysis). If significant differences exist in these individuals, a bias may be introduced. For instance, individuals around the age of the full sample mean (~55 years) may be starting to see the detrimental health impacts of long-term smoking and be enticed to quit during the sampling period. The individuals most impacted by smoking may be the ones most likely to quit, thus introducing a potential bias by their exclusion. Individuals that can smoke into old-age and stay reasonably healthy while doing so (due to protective factors, genetics, etc.) might be overrepresented in this sample. The cross-sectional relationship between smoking and LTL is typically very small and could easily be overwhelmed by such biases.

4. Methodological: A key feature of Hill’s (1965) seminal work on causality in epidemiological research is biological gradient, wherein increased exposure results in the increased likelihood and severity of the outcome. The authors might benefit from the inclusion of information about dose response (i.e., pack-year of smoking, as in Astuti et al. 2017) into the manuscript and how it might impact the results.

5. Methodological: Three cohorts, ADE (2 vs. 42), ERA (1 vs. 86), & LBC1921 (3 vs. 76), had overwhelmingly unbalanced distributions of smokers and nonsmokers. These studies are severely underpowered to make claims about differences in LTL between smokers and non-smokers. This limitation is not discussed, despite its impacts on several key aspects of the study highlighted below. The manuscript could be improved by adding supplementary analyses excluding these three studies.

a. Two of these studies (ADE & ERA) comprise half of the studies using the TRF method, estimates of which were used to convert SMDs to units of base pairs. While this conversion may be defensible as described based on strengths of the measurement method used (e.g. TRF relative to qPCR), taking standard deviation estimates of LTL and LTL attrition from cohorts containing only one to three smokers and applying these estimates to SMDs from other studies with over 1,000 smokers seems inappropriate.

b. The same two cohorts (ADE & ERA) also had high correlation between baseline and follow-up measurements. Thus, the authors' decision to rerun model 6, "weighting the contribution of each cohort by the correlation between baseline and follow-up LTL measurements" (page 9), grossly overvalues the contribution of these studies toward the overall effects observed across all 18 studies.

6. Analysis: Was the impact of chronological age on LTL controlled for in the analyses investigating differences in LTL between smokers and non-smokers at baseline and follow-up (i.e. Models 2 and 3)? The only models in which chronological age are explicitly stated to be included are Models 1 and 5.

7. Analysis: Age-adjusted LTL measures should similarly be used to recalculate LTL attrition rates. As currently described, LTL attrition rates were calculated without controlling for any factors at baseline, including baseline LTL. The latter was excluded because, "the observed correlation between baseline LTL and attrition rate is regression to the mean resulting largely from measurement error" (pages 10-11). This statistical rationale is tenable. However, other control variables exist as the authors themselves acknowledged on page 12: "controlling for baseline telomere length in multiple regression models of telomere attrition biases the effects of other predictor variables that correlate with telomere length at baseline like age, sex, and smoking status." However, the authors do not go on to account for these variables in their models. In other words, I wonder whether the effects of these other variables are exaggerated because they are not considered at all? I suggest the authors reconsider their adjustment for age and sex at baseline and follow-up in their analyses of LTL attrition rates.

Miscellaneous comments:

- In describing their literature search, the authors note the use of snowballing. For snowballing to be done adequately, the set of papers from which the authors attempted to move forward and backward through needed to be carefully vetted. Including the PRISMA flow diagram (how many papers screened etc.) will improve the transparency of the data presentation.

-There is no limitation section included in the discussion. Some of the topics highlighted above can be included in this section (i.e., selective attrition of smokers, potential sex differences, SES, dose-response etc.) Doing so would help transform the otherwise (implied) causal language used to support the selective adoption hypothesis into more appropriate tentative conclusions.

Idan Shalev
Penn State University

Author's Response to Decision Letter for (RSOS-181763.R0)

See Appendix A.

RSOS-190420.R0

Review form: Reviewer 1 (Maya Mathur)

Is the manuscript scientifically sound in its present form?

Yes

Are the interpretations and conclusions justified by the results?

Yes

Is the language acceptable?

Yes

Is it clear how to access all supporting data?

Yes

Do you have any ethical concerns with this paper?

No

Have you any concerns about statistical analyses in this paper?

No

Recommendation?

Accept as is

Comments to the Author(s)

I thank the authors for their thorough responses to my comments. I continue to feel this is a strong paper and am pleased to recommend acceptance at this point.

Review form: Reviewer 2 (Idan Shalev)

Is the manuscript scientifically sound in its present form?

Yes

Are the interpretations and conclusions justified by the results?

Yes

Is the language acceptable?

Yes

Is it clear how to access all supporting data?

Yes

Do you have any ethical concerns with this paper?

No

Have you any concerns about statistical analyses in this paper?

No

Recommendation?

Accept with minor revision (please list in comments)

Comments to the Author(s)

I appreciate the authors' response to my comments. The manuscript has been revised in accordance with my suggestions. I believe the findings support the authors' conclusions and can contribute significantly to the literature on the effects of smoking on telomere attrition.

I have one remaining suggestion. While re-reading and reflecting on the manuscript I realized that perhaps the authors can begin and end their manuscript by pointing to the overall damaging effects of smoking on health. While they have effectively shown that smoking does not accelerate the rate of telomere attrition and that selective adoption is a plausible explanation, it should be clarified that their results do not preclude the myriad of other damaging effects of smoking that are well established by the literature. I mention this because while I trust the author's themselves have no conflict of interest, ethically it is key that they help mitigate potential exploitative uses of their findings by the tobacco industry and its stakeholders.

Decision letter (RSOS-190420.R0)

11-Apr-2019

Dear Dr Bateson

On behalf of the Editor, I am pleased to inform you that your Manuscript RSOS-190420 entitled "Smoking does not accelerate leukocyte telomere attrition: a meta-analysis of 18 longitudinal cohorts" has been accepted for publication in Royal Society Open Science subject to minor revision in accordance with the referee suggestions. Please find the referees' comments at the end of this email.

The reviewers and Subject Editor have recommended publication, but also suggest some minor revisions to your manuscript. Therefore, I invite you to respond to the comments and revise your manuscript.

- Ethics statement

- Data accessibility

It is a condition of publication that all supporting data are made available either as supplementary information or preferably in a suitable permanent repository. The data

accessibility section should state where the article's supporting data can be accessed. This section should also include details, where possible of where to access other relevant research materials such as statistical tools, protocols, software etc can be accessed. If the data has been deposited in an external repository this section should list the database, accession number and link to the DOI for all data from the article that has been made publicly available. Data sets that have been deposited in an external repository and have a DOI should also be appropriately cited in the manuscript and included in the reference list.

If you wish to submit your supporting data or code to Dryad (<http://datadryad.org/>), or modify your current submission to dryad, please use the following link:
<http://datadryad.org/submit?journalID=RSOS&manu=RSOS-190420>

- **Competing interests**

- **Authors' contributions**

- **Acknowledgements**

- **Funding statement**

Because the schedule for publication is very tight, it is a condition of publication that you submit the revised version of your manuscript before 20-Apr-2019. Please note that the revision deadline will expire at 00.00am on this date. If you do not think you will be able to meet this date please let me know immediately.

To revise your manuscript, log into <https://mc.manuscriptcentral.com/rsos> and enter your Author Centre, where you will find your manuscript title listed under "Manuscripts with Decisions". Under "Actions," click on "Create a Revision." You will be unable to make your

revisions on the originally submitted version of the manuscript. Instead, revise your manuscript and upload a new version through your Author Centre.

on behalf of Dr John Dalton (Associate Editor) and Professor Kevin Padian (Subject Editor)
openscience@royalsociety.org

Associate Editor Comments to Author (Dr John Dalton):

The manuscript has received a good responses from reviewers. One final potential important issue was raised by one of the reviewers which the authors could consider

"While they have effectively shown that smoking does not accelerate the rate of telomere attrition and that selective adoption is a plausible explanation, it should be clarified that their results do not preclude the myriad of other damaging effects of smoking that are well established by the literature."

Reviewer comments to Author:

Reviewer: 1

Comments to the Author(s)

I thank the authors for their thorough responses to my comments. I continue to feel this is a strong paper and am pleased to recommend acceptance at this point.

Reviewer: 2

Comments to the Author(s)

I appreciate the authors' response to my comments. The manuscript has been revised in accordance with my suggestions. I believe the findings support the authors' conclusions and can contribute significantly to the literature on the effects of smoking on telomere attrition.

I have one remaining suggestion. While re-reading and reflecting on the manuscript I realized that perhaps the authors can begin and end their manuscript by pointing to the overall damaging effects of smoking on health. While they have effectively shown that smoking does not accelerate the rate of telomere attrition and that selective adoption is a plausible explanation, it should be clarified that their results do not preclude the myriad of other damaging effects of smoking that are well established by the literature. I mention this because while I trust the author's themselves have no conflict of interest, ethically it is key that they help mitigate potential exploitative uses of their findings by the tobacco industry and its stakeholders.

Author's Response to Decision Letter for (RSOS-190420.R0)

See Appendix B.

Decision letter (RSOS-190420.R1)

03-May-2019

Dear Dr Bateson,

I am pleased to inform you that your manuscript entitled "Smoking does not accelerate leukocyte telomere attrition: a meta-analysis of 18 longitudinal cohorts" is now accepted for publication in Royal Society Open Science.

Royal Society Open Science operates under a continuous publication model (<http://bit.ly/cpFAQ>). Your article will be published straight into the next open issue and this will be the final version of the paper. As such, it can be cited immediately by other researchers.

As the issue version of your paper will be the only version to be published I would advise you to check your proofs thoroughly as changes cannot be made once the paper is published.

on behalf of Dr John Dalton (Associate Editor) and Kevin Padian (Subject Editor)
openscience@royalsociety.org

Associate Editor Comments to Author (Dr John Dalton):
Associate Editor: 1
Comments to the Author:
(There are no comments.)

Reviewer comments to Author:

Appendix A

Dear editors,

Thank you for the helpful comments on our paper. We have now fully revised the paper in the light of the referees' comments and feel the paper is substantially stronger as a result of the changes. None of the additional analyses that we have conducted in response to the referees' comments have led to any qualitative changes in the results we present. Below we detail our response to each of the referees' comments and how we have changed the paper as a result.

Yours faithfully,

Melissa Bateson.

Editor Comments:

We have somewhat divergent reviews on this MS, both of them thoughtful. Dr. Shalev in particular brings up some important issues for consideration that may require re-analysis and discussion among the (many) authors. Our timeframe for a simple "revise" decision is short and so mainly for this reason I log a "reject/resubmi" recommendation. However it will be important to address directly all of Dr. Shalev's comments as well as those of the other reviewer. Thanks for submitting.

Reviewers' Comments to Author:

Reviewer: 1

Comments to the Author(s)

OVERALL COMMENTS

This exemplary meta-analysis was a pleasure to read. The design and statistical methods evidence careful attention; I was pleased with the use of broad search terms likely to capture all relevant published literature, the inclusion of only longitudinal studies with adequate follow-up, the use of directly comparable statistical estimates from each study (which often required contacting the original authors or re-analyzing published data), and the sensible, thorough statistical reporting. The Discussion and sensitivity analyses show careful consideration of possible alternative explanations. I furthermore confirm that the authors have publicly provided well-documented, clear code and data to reproduce their results.

I therefore believe that the primary finding that smoking is not longitudinally associated with telomere attrition is an excellent example of a persuasively null result. I would be happy to see the paper accepted in its present form, as I have only the following minor suggestions.

MINOR COMMENTS

1.) Methods: Were there any differences in the analyses conducted in studies in which the authors had access to individual participant data (IPD) vs. only summary statistics? For example, the penultimate paragraph of "Methods" describes subject-level exclusion criteria. Were these same criteria applied when only summary estimates were available?

Response: For the nine cohorts for which we used summary statistics supplied by co-authors, the first author (MB) provided detailed instructions on the statistics required and how these should be calculated. These instructions replicated the approach taken with the seven cohorts for which she had full individual participant data available to make the analyses as consistent as possible across

the 18 cohorts. We have added the following sentence to the methods to clarify this point: ‘In the cases where we requested summary statistics, we provided detailed instructions on how these were to be calculated (including any exclusions – see below) in order to ensure consistency across cohorts in how the data were analysed’.

2.) Methods: Given that IPD were available for 9 cohorts, would it have been possible to conduct counterpart subject-level analyses on this subset of the data? Was this infeasible due to different measurement scales for the LTL variables?

Response: It is feasible (although practically not very straightforward) to conduct a subject-level analysis for the subset of cohorts for which we have IPD (but, note that there were only 7 cohorts for which we had IPD, not 9). We were intrigued to understand whether this latter approach would yield a different result from meta-analysis. To answer this question, we focussed on models of LTL attrition (e.g. model 6), since this is the critical analysis in our paper. We used simulated data to establish that for data like ours, where there is non-significant heterogeneity in the effect of smoking on LTL attrition between cohorts, a subject-level analysis (in which the subject-level data from multiple cohorts are pooled, with or without a random effect of cohort) is essentially equivalent to a fixed-effects meta-analysis of the same dataset: parameter estimates and p-values are nearly identical. This is important, because it means that the question asked by the referee can be re-stated as asking: what is the difference between a fixed and random-effects meta-analysis? This question can be answered using all 18 cohorts rather than having to restrict ourselves to the subset of seven cohorts for which we have IPD. The effect of moving from a random-effects meta-analysis to a fixed-effects meta-analysis should be that cohorts with more extreme effect sizes will gain influence if they are large and lose influence if they are small (i.e. the weighting of cohorts will be less balanced under a fixed-effects approach). We re-ran model 6 with a fixed-effects meta-analysis to see how much difference this made. The summary parameter estimate for the association between smoking and LTL attrition was even closer to zero than with the random-effects model (parameter estimate and 95% CI: -0.01 [-0.05, 0.02], $p = 0.4434$). Thus, it appears to make no difference to our general conclusion which model we use: both models yield parameter estimates for the effect of smoking on LTL attrition not significantly different from zero. While it is feasible to fit a fixed-effects meta-analysis, we do not believe that it makes sense theoretically due to the considerable variation between the cohorts in age, BMI, nationality, ethnicity and the methods used to measure LTL. Indeed, our rationale for originally choosing random-effects over fixed-effects meta-analysis models in the paper was because we deemed the cohorts too different to justify assuming that there are common true effect sizes to estimate (for the same reason, we would not wish to base our paper on subject-level analysis of data pooled from different cohorts, even if this were straightforward for all 18). Thus, we have retained the original random-effects meta-analysis in the paper on the basis of this a priori justification. The fixed-effects analysis of model 6 is included in the R script for completeness. We have added a sentence to the statistical methods to justify the use of random-effects meta-analysis.

3.) Methods, final paragraph: I initially wasn't sure what "pooled LTL" referred to, though I eventually inferred that it meant "pooled across time points".

Response: We have clarified this point.

4.) Results, final paragraph, and page 9: I am not sure that variation in the correlation between baseline and F/U LTL is necessarily evidence for measurement error. Couldn't it simply be that some cohorts have more inherent variation in LTL attrition than others, for example due to demographic or behavioral characteristics?

Response: The referee is correct that heterogeneity between cohorts in the variation in LTL attrition within cohorts could affect the correlation between baseline and follow-up LTL for a

cohort. However, there is strong evidence to suggest that much of the variation in this correlation arises from measurement error, because the correlation is strongly associated with TL measurement method: datasets measured with more precise TRF always have higher correlations than those measured with less precise qPCR (this is very clearly true in the current dataset – see the correlations in Table S1). We have added a justification for the use of the correlation as a proxy for measurement error to the statistical methods section: ‘In one of our sensitivity analyses we instead weighted cohorts by the correlation between baseline and follow-up LTL measurements (r). The rationale for this decision is that r has been argued to be a good proxy for LTL measurement error (Nettle et al. 2018; Steenstrup et al. 2013), an assumption supported by the observation that datasets measured with more precise TRF always have higher correlations than those measured with less precise qPCR (as shown by the values in Table S1).’

5.) Page 12, “While there is no data relating specifically to genes linked to behavior”: Do you mean telomeres rather than “genes” broadly speaking? Of course genes are linked to behavior.

Response: This sentence was badly worded. We have re-written it as follows: ‘There is emerging evidence that telomere shortening causes changes in regulation of more than 140 genes (Robin et al. 2014; Kim et al. 2016), making this idea theoretically possible (although none of the genes identified thus far as regulated by telomere shortening is obviously linked to behaviour, let alone smoking)’.

6.) Very last paragraph: Do you perhaps mean “selective adoption” instead of “selective causation”?

Response: Yes, thanks. This was a mistake that has now been corrected.

Reviewer: 2

Comments to the Author(s)

The authors present a meta-analysis of 18 cohort studies with longitudinal data on telomere length and smoking status. Prevailing thinking assumes a causal link exists between smoking and accelerated telomere shortening. However, this claim is based largely on cross-sectional data. The current manuscript provocatively challenges this view, presenting a fresh and alternative hypothesis of ‘selective adoption’ that assumes individuals with shorter telomere length are more likely to start smoking, and that some other factors both shorten telomeres and predispose individuals toward smoking behavior. The paper is well-written and theoretically sound. However, the manuscript can be improved prior to publication by addressing several conceptual, inferential and methodological issues.

1. Inferential: The current work presents 3 key findings: a) LTL in smokers is shorter than LTL in non-smokers at baseline and follow up, b) LTL decreases over the span of time between baseline and follow up, and c) the rate of LTL attrition between baseline and follow-up is not significantly different in smokers relative to non-smokers. These findings are explained in the context of the ‘causal’ and ‘selective adoption’ hypotheses for the effect of smoking on telomere length. However, the results of this meta-analysis neither prove nor disprove the causal or selective adoption hypotheses since no LTL measurements exist prior to the onset of smoking. For the most part, the authors are careful in their wording of what conclusions can be derived from their work. However, there are a few places where uses of strong language imply that the data completely support the selective adoption hypothesis. Overall, the authors are recommended to use a more suggestive language by removing any (implied) causal inferences.

For example, page 12: “Our findings do not support the hypothesis that smoking is the sole cause of differences in LTL between smokers and non-smokers. Accordingly, we conclude that selective adoption of smoking by individuals with short telomeres is likely to be occurring”. The data show shorter LTL in smokers versus non-smokers across two time points, and thus, provides no evidence for what may be causing the difference. It could be smoking if the effect of smoking had completely manifested before baseline LTL measurements. In fact, smoking may very well be one of the causes since it is the only variable different in these individuals at baseline other than their LTL (since nothing else was controlled for). Furthermore, beginning the next sentence with “accordingly” implies an “if not A then B” logic to the relationship. In other words, the passage implies that if LTL is not wholly shortened by smoking then selective adoption must be happening.

Response: We respectfully disagree with the statement that that ‘the results of this meta-analysis neither prove nor disprove the causal or selective adoption hypotheses since no LTL measurements exist prior to the onset of smoking’, as long as the ‘causal hypothesis’ is taken to mean the causation hypothesis as we set it out in Figure 1.

The aim of Figure 1 is to make it crystal clear that it is a unique prediction of the selective adoption hypothesis that the rate of LTL attrition will be equal in smokers and non-smokers. Thus, demonstrating that rates of LTL attrition are no different in smokers and non-smokers is sufficient to disprove the causation and mixed hypotheses as we have defined them here. In contrast, the existence of a difference in LTL prior to the onset of smoking is compatible with both the selective adoption and the mixed hypothesis. Thus, although such a difference would be compatible with selective adoption, it is not in itself sufficient to prove that causation (as we define it) is not also occurring. There are other possible hypotheses not in the set we outline in Figure 1, and we do explore these in our analyses and acknowledge them in the discussion (though the evidence seems to make them unlikely, see below).

While our current analyses conclusively rule out the causation and mixed hypotheses depicted in Figure 1, whereby smoking causes a sustained increase in the rate of LTL attrition, we concur with the referee that our analyses do not completely rule out a highly non-linear effect, whereby smoking causes a short-term acceleration of telomere attrition that rapidly normalises. We addressed this possibility by examining whether the rate of LTL attrition is predicted by the age of the cohort and we found no evidence that the rate of attrition in smokers decreases with cohort age as would be predicted under such a non-linear scenario (Fig 3b). We agree that to decisively rule out a non-linear effect that is completely restricted to the decade immediately after starting smoking we need to know whether telomere length differences in childhood precede the initiation of smoking, as the referee says. We discuss these points in the second paragraph of the discussion.

We have reviewed all of our conclusions carefully to make sure that our statements follow directly from our findings and have made the following changes to address the referee’s comments. First, we have modified the end of the second paragraph of the introduction to read: ‘Our aim in the current paper is therefore to use a meta-analysis of longitudinal LTL attrition data to directly test the hypothesis that smoking causes a sustained increase in the rate of LTL attrition in adults (henceforth the causation hypothesis). We additionally propose an alternative hypothesis of selective adoption, whereby individuals with shorter LTL are more likely to start smoking. Second, we have modified the last sentence of the first paragraph of the discussion to read: ‘Taken together, these findings provide no support for the hypothesis that smoking causes a sustained increase in the rate of LTL attrition.’ Third, we have deleted the paragraph referred to by the referee starting, ‘Our findings do not support the hypothesis that smoking is the sole cause of differences in LTL between smokers and non-smokers...’ and replaced it with the sentence: ‘In the absence of any evidence supporting the hypothesis that smoking causes a sustained increase in

the rate of LTL attrition, it is worth considering the alternative hypothesis that selective adoption is occurring...'. Finally, we have rewritten the final sentence of the abstract to read: 'Therefore, the difference in LTL between smokers and non-smokers is extremely unlikely to be explained by a linear causal effect of smoking. Selective adoption, whereby individuals with short telomeres are more likely to start smoking, needs to be considered as a more plausible explanation for the observed pattern of telomere dynamics.'

2. Conceptual: While the 'selective adoption' hypothesis is interesting, and plausible, perhaps the more parsimonious explanation for the 'selective adoption' could simply be a confounding variable? Whether it is, for example, prior exposure to early-life adversity, low SES, health status, alcohol use, or just education, specifically. For example, when controlling for SES, the effect of smoking is usually reduced. Further, the potential relationship between LTL and smoking through early-life adversity could potentially be tested using other datasets such as the UK Environmental Risk cohort with repeated telomere length measurements prior to smoking at ages 5, 10 and 18 years. This study, which includes variables such as exposure to violence and telomere length prior to smoking initiation and smoking status at age 18 years, can be used to test the authors' 'selective adoption' hypothesis. **Response: In response to the first point, we completely agree. We use 'selective adoption' as a blanket term to cover both reverse causation (telomere length causes smoking) and third-variable explanations (something else—such as genes or early-life experience--causes both telomere length and smoking), because in both cases individuals with shorter telomeres will be more likely to start smoking (see Figure 1 in Bateson and Nettle 2018).**

We also completely agree with the second point. We have made this point in a previous paper (Bateson and Nettle 2018) and make it again in the second paragraph of the discussion of the current paper. Since writing this paper, various cohorts have become available in which it might be possible to test the prediction of selective adoption that differences in LTL will precede starting smoking. We are currently actively pursuing this line of investigation and the results will be reported in a future meta-analysis.

3. Methodological: The argument for the exclusion of the approximately 1,200 people that had ambiguous smoking status would be strengthened with descriptive statistics for this group (e.g. age, smoking status change, and other variables common between the studies included in this meta-analysis). If significant differences exist in these individuals, a bias may be introduced. For instance, individuals around the age of the full sample mean (~55 years) may be starting to see the detrimental health impacts of long-term smoking and be enticed to quit during the sampling period. The individuals most impacted by smoking may be the ones most likely to quit, thus introducing a potential bias by their exclusion. Individuals that can smoke into old-age and stay reasonably healthy while doing so (due to protective factors, genetics, etc.) might be overrepresented in this sample. The cross-sectional relationship between smoking and LTL is typically very small and could easily be overwhelmed by such biases.

Response: We were concerned about the potential for selection bias of the type described here. However, we are not convinced that comparing participants that were included with those that were excluded due to inconsistent smoking status (or death) is the best way to address this issue. We explored the possibility that those participants retained in the study were more resistant to the long-term effects of smoking via two additional analyses reported in the results section. The argument and results are described in the following passage in the discussion: 'by selecting against individuals who die between baseline and follow-up, longitudinal studies could underestimate the effects of smoking on telomere attrition, because they retain only smokers who are resistant to the damaging effects of tobacco smoke. However, this argument fails to explain the substantial difference in baseline LTL between smokers and non-smokers that we observed, even in the studies where the age of participants at baseline was quite advanced. If selection bias is present, it should affect cross-sectional associations as well as measures of attrition, yet we found no change

in the difference in LTL between smokers and non-smokers with increasing cohort age (Fig. 3b). Furthermore, selection based on mortality should be negligible in cohorts in their 20s or 30s, but substantial for cohorts over 60, yet we found no effect of baseline age on the size of the association between smoking and LTL attrition, despite an age range of over 54 years’.

4. Methodological: A key feature of Hill’s (1965) seminal work on causality in epidemiological research is biological gradient, wherein increased exposure results in the increased likelihood and severity of the outcome. The authors might benefit from the inclusion of information about dose response (i.e., pack-year of smoking, as in Astuti et al. 2017) into the manuscript and how it might impact the results.

Response: In meta-analysis there is generally a trade-off between the sensitivity with which it is possible to characterise a variable and the number of studies it is possible to include, because not all studies capture all variables with the same level of detail. We opted to classify subjects as smokers or non-smokers because this information was available for all 18 cohorts. More detailed information on dose of smoking was only available for a subset of cohorts and different cohorts used different categories making it impossible to combine data. We have added the following sentence to the methods: ‘We did not attempt to explore the effects of amount smoked, since consistent data were not available for all cohorts’. Furthermore, even if it had been possible to use dose information in our analyses, this would not have helped sort out causality, because data exist showing that early-life adversity predicts not only future smoking, but also the amount smoked (Anda et al. 1999). Thus both a causal hypothesis and a third-variable hypothesis are compatible with an association between amount smoked and TL. We have added a couple of sentences to the discussion to make this point: ‘It is worth noting that, the available data lead us to predict not only an association between childhood LTL and the presence of adult smoking, but also between childhood LTL and the amount smoked. An association between LTL and amount smoked is often regarded as strong evidence for the causation hypothesis (Astuti et al. 2017). However, it is now clear that such evidence is equally compatible with selective adoption.’

5. Methodological: Three cohorts, ADE (2 vs. 42), ERA (1 vs. 86), & LBC1921 (3 vs. 76), had overwhelmingly unbalanced distributions of smokers and nonsmokers. These studies are severely underpowered to make claims about differences in LTL between smokers and non-smokers. This limitation is not discussed, despite its impacts on several key aspects of the study highlighted below. The manuscript could be improved by adding supplementary analyses excluding these three studies.

Response: We agree that the ADE, ERA and LBC1921 cohorts have very low power due to the low numbers of smokers (this is evident in the huge confidence intervals for these cohorts in the forest plots). We did not know the numbers of smokers at the time of requesting these data and it would have been arbitrary to exclude them post-hoc on the basis of sample size. However, it should not matter that these cohorts have low power, because the point of meta-analysis is to combine estimates of effect size from individual studies that might themselves be underpowered in order to obtain higher overall power. The meta-analysis model we used weights the contributions of studies using an inverse-variance algorithm, whereby cohorts such as these end up contributing relatively little to the summary effect size estimate. Thus our analyses already control for the low power in these three cohorts. We therefore feel that it is both unnecessary and undesirable to exclude them (note that we already have a leave-one-out sensitivity analysis in which we have shown that our results are robust to the exclusion of any single cohort). To reassure the referee that our results are robust to excluding the three very low-powered cohorts we have re-run model 6 (the critical model of rate of attrition) without them. Exactly as we would have predicted, the results are effectively unchanged by excluding the three low-powered cohorts (parameter estimate for model 6 and 95% CI = -0.02 [-0.08, 0.04], $p = 0.4966$).

a. Two of these studies (ADE & ERA) comprise half of the studies using the TRF method, estimates of which were used to convert SMDs to units of base pairs. While this conversion may be defensible as described based on strengths of the measurement method used (e.g. TRF relative to qPCR), taking standard deviation estimates of LTL and LTL attrition from cohorts containing only one to three smokers and applying these estimates to SMDs from other studies with over 1,000 smokers seems inappropriate.

Response: This concern is based on a misunderstanding of the conversion. In order to convert standardised parameter estimates into base pairs we needed an estimate of the standard deviation (of either LTL or attrition, depending on the model). The sd required is not for smokers and non-smokers separately, but for the cohort as a whole, pooling the data from smokers and non-smokers. Therefore, the precision of the estimates of sd depend on the total n for each cohort. The ns for the four TRF cohorts are: 44, 635, 87 and 429 for ADE, BHS, ERA and JLRCs respectively. In the original version of the paper we used the mean of the sds from these four cohorts in our calculations. However, on reflection, we have realised that this approach is not optimal. A simple, unweighted mean is not the best estimate of sd, because larger cohorts will yield more precise estimates of sd than small cohorts and the precision of an estimate grows with the square root of the sample size. Thus, we have recalculated the sds used for conversion with a weighted mean in which we weight the sds from each cohort according the square root of n for the cohort. This new approach gives less weight to the sds derived from the two smallest cohorts (ADE & ERA). While the resulting sds are slightly different from those reported in the original paper, leading to small differences in the calculated LTL and attrition estimates, there is no change to any of our conclusions.

b. The same two cohorts (ADE & ERA) also had high correlation between baseline and follow-up measurements. Thus, the authors' decision to rerun model 6, "weighting the contribution of each cohort by the correlation between baseline and follow-up LTL measurements" (page 9), grossly overvalues the contribution of these studies toward the overall effects observed across all 18 studies.

Response: It is important to note that the headline results that we present in Table 1, the discussion and abstract are based on model 6 run with the conventional inverse-variance weighting of cohorts. The re-running of model 6 weighting by the correlation between baseline and follow-up was a subsidiary sensitivity analysis to appease those who believe that the high measurement error present in some qPCR measurements devalues these data. In fact, it turned out to make no significant difference how we weighted the cohort contributions in model 6 (as we report in the results section), so this is really not an issue.

6. Analysis: Was the impact of chronological age on LTL controlled for in the analyses investigating differences in LTL between smokers and non-smokers at baseline and follow-up (i.e. Models 2 and 3)? The only models in which chronological age are explicitly stated to be included are Models 1 and 5.

Response: The referee is correct that we did not control for chronological age in models 2 and 3. The logic for our sequence of analyses for the cross-sectional data was as follows. First, we asked whether there was an association between smoking and LTL at baseline and follow-up (models 2 and 3 respectively). Since we found significant associations in both cases, we then asked whether the association was stronger at follow-up (as predicted by the causation and mixed hypotheses; model 4). Since there was no difference between baseline and follow-up, we combined the data from baseline and follow-up to obtain a better estimate of the association between smoking and LTL. As expected, there was a significant association between smoking and LTL using this combined measure of LTL (model 5). Finally, we asked whether the heterogeneity between cohorts present in model 5 could be explained by adding age as a moderator to model 5. We have endeavoured to convey this logic clearly in the results section. To establish whether there was any

value to including additional analyses in which we explored the effect of age as a moderator in models 2 and 3 we ran these analyses. In doing this, we realised that it makes sense to use slightly different values of age in the three analyses: mean age at baseline for model 2, mean age at follow-up for model 3 and mean age at the mid-point of the cohort for model 5 (in the previous version of the paper we had used age at baseline in model 5). The consequences of adding age to models 2, 3 and 5 are shown in the table below. The results are extremely similar, and we believe that there is little to be gained by adding the additional analyses for models 2 and 3 to the paper. However, we have added these analyses in the R script for completeness. We have also updated the results presented for model 5 in the paper to reflect the use of mean age at the mid-point as opposed to mean age at baseline; this makes no difference to the original conclusion that there is no association between age and the effect of smoking on LTL.

Model to which age is added	Age used	Parameter estimate (= slope of meta-regression)	95% CI	p-value
Model 2 (baseline LTL)	Mean age at baseline	0.00	[-0.00, 0.00]	0.9542
Model 3 (follow-up LTL)	Mean age at follow-up	0.00	[-0.01, 0.01]	0.9948
Model 5 (combined LTL)	Mean age at mid-point of follow-up	0.00	[-0.01, 0.01]	0.9450

7. Analysis: Age-adjusted LTL measures should similarly be used to recalculate LTL attrition rates. As currently described, LTL attrition rates were calculated without controlling for any factors at baseline, including baseline LTL. The latter was excluded because, “the observed correlation between baseline LTL and attrition rate is regression to the mean resulting largely from measurement error” (pages 10-11). This statistical rationale is tenable. However, other control variables exist as the authors themselves acknowledged on page 12: “controlling for baseline telomere length in multiple regression models of telomere attrition biases the effects of other predictor variables that correlate with telomere length at baseline like age, sex, and smoking status.” However, the authors do not go on to account for these variables in their models. In other words, I wonder whether the effects of these other variables are exaggerated because they are not considered at all? I suggest the authors reconsider their adjustment for age and sex at baseline and follow-up in their analyses of LTL attrition rates.

Response: In designing this meta-analysis we carefully considered whether we should use estimates of the effect of smoking on telomere attrition that controlled for other variables such as baseline TL, age and sex. Given the evidence that estimates of the effects of exposures (including age, sex and smoking) on telomere attrition are likely to be overestimated in models that control for baseline TL (Bateson, Eisenberg, and Nettle 2018; Needham et al. 2019), we elected to use unadjusted estimates of the effect of smoking on telomere attrition in the current study to avoid this problem. We are therefore unable to control for age or sex in the current analysis, because controlling for these variables would require fitting new, multiple regression models (with age and sex but not baseline LTL) to the data from each cohort and we do not have the individual participant data available for the majority of the cohorts that would be necessary to do this. Given that we cannot control for age and sex with the data we have, the critical question is whether there is any evidence that our decision not to control for these variables could affect the results of the current meta-analysis?

Although we did not control for age and sex in the current paper, the majority of the nine published studies that we have been able to find reporting effects of smoking on telomere attrition report the results of multiple regression models that do control for age and sex (Aviv et

al. 2009; Bendix et al. 2014; Ehrlenbach et al. 2009; Farzaneh-Far et al. 2010; Huzen et al. 2014; Müezziner et al. 2015; Révész et al. 2016; Toupance et al. 2017; Weischer, Bojesen, and Nordestgaard 2014). **Considered together, the results of these nine studies support the general conclusions of the current paper: there is strong evidence for an effect of smoking on LTL in cross-sectional analyses (six out of eight studies that tested for a difference report that LTL is significantly shorter in smokers), but there is much less evidence for an effect of smoking on telomere attrition in longitudinal analyses (only two out of nine studies report that telomere attrition is significantly faster in smokers). We have already discussed this evidence elsewhere (see Table 3 in Bateson and Nettle 2018).**

Furthermore, it is reassuring that our meta-analysis of the cross-sectional effect of smoking on LTL produces a summary effect size for smoking (-0.13) that is similar to that reported in another meta-analysis based on published effect sizes derived from cross-sectional studies that do control for potential confounds such as age and sex (-0.11 ; Astuti et al. 2017). Thus, the conclusions drawn from analyses that do and don't control for age and sex appear very similar. We therefore believe that there is no evidence to suggest that controlling for age and sex would alter the conclusions from our meta-analysis. We have added a paragraph summarising the above arguments to the discussion of the paper.

As a final point, our response to this comment thus far has assumed that the referee is referring to statistical control at the individual participant level within each cohort. However, some of his language (e.g. 'the authors do not go on to account for these variables in their models') implies that he would like us to control for age and sex at the meta-analytic level (since these are the only models we fit in the paper). For this to make any sense there needs to be variation between cohorts. In the case of age, there is substantial variation in the mean age of cohorts and we have already tested various hypotheses with respect to age at the meta-analytic level (e.g. adding age to model 5 to ask whether the effect of smoking on LTL increases with age and adding age to model 6 to ask whether the effect of smoking on LTL attrition declines with age).

Miscellaneous comments:

- In describing their literature search, the authors note the use of snowballing. For snowballing to be done adequately, the set of papers from which the authors attempted to move forward and backward through needed to be carefully vetted. Including the PRISMA flow diagram (how many papers screened etc.) will improve the transparency of the data presentation.

Response: We have included a PRISMA diagram as a new Figure S1 that clarifies the number of cohorts obtained from different sources.

-There is no limitation section included in the discussion. Some of the topics highlighted above can be included in this section (i.e., selective attrition of smokers, potential sex differences, SES, dose-response etc.) Doing so would help transform the otherwise (implied) causal language used to support the selective adoption hypothesis into more appropriate tentative conclusions.

Response: Although we do not have a single limitations section, we do address in considerable depth several possible limitations of our study. As requested, there are paragraphs addressing both the potential impact of selective attrition of smokers and of age and sex differences (see our response to the earlier comment). We cannot say anything about SES since this is not quantified or reported in the majority of cohorts. We have addressed the question of dose-response in our response to an earlier comment.

References cited in response to referees:

Anda, Robert F, Janet B Croft, Vincent J Felitti, Dale Nordenberg, Wayne H Giles, David F Williamson, Gary A Giovino, and Patient Page. 1999. "Adverse Childhood Experiences and Smoking during

- Adolescence and Adulthood." *JAMA* 282: 1652–58.
- Astuti, Yuliana, Ardyan Wardhana, Johnathan Watkins, and Wahyu Wulaningsih. 2017. "Cigarette Smoking and Telomere Length: A Systematic Review of 84 Studies and Meta-Analysis." *Environmental Research* 158: 480–89. <https://doi.org/10.1016/j.envres.2017.06.038>.
- Aviv, Abraham, Wei Chen, Jeffrey P. Gardner, Masayuki Kimura, Michael Brimacombe, Xiaojian Cao, Sathanur R. Srinivasan, and Gerald S. Berenson. 2009. "Leukocyte Telomere Dynamics: Longitudinal Findings among Young Adults in the Bogalusa Heart Study." *American Journal of Epidemiology* 169 (3): 323–29. <https://doi.org/10.1093/aje/kwn338>.
- Bateson, Melissa, Dan T.A. Eisenberg, and Daniel Nettle. 2018. "Controlling for Baseline Telomere Length Biases Estimates of the Rate of Telomere Attrition." *Zenodo*, <http://doi.org/10.5281/zenodo.1009086>. <https://doi.org/10.5281/zenodo.1009086>.
- Bateson, Melissa, and Daniel Nettle. 2018. "Why Are There Associations between Telomere Length and Behaviour?" *Philosophical Transactions of the Royal Society of London. Series B, Biological Sciences* 373: 20160438. <https://doi.org/10.1098/rstb.2016.0438>.
- Bendix, Laila, Mikael Thinggaard, Mogens Fenger, Steen Kolvraa, Kirsten Avlund, Allan Linneberg, and Merete Osler. 2014. "Longitudinal Changes in Leukocyte Telomere Length and Mortality in Humans." *Journals of Gerontology - Series A Biological Sciences and Medical Sciences* 69 A (2): 231–39. <https://doi.org/10.1093/gerona/glt153>.
- Ehrlebenbach, Silvia, Peter Willeit, Stefan Kiechl, Johann Willeit, Markus Reindl, Kathrin Schanda, Florian Kronenberg, and Anita Brandstätter. 2009. "Influences on the Reduction of Relative Telomere Length over 10 Years in the Population-Based Bruneck Study: Introduction of a Well-Controlled High-Throughput Assay." *International Journal of Epidemiology* 38 (6): 1725–34. <https://doi.org/10.1093/ije/dyp273>.
- Farzaneh-Far, Ramin, Jue Lin, Elissa Epel, Kyle Lapham, Elizabeth Blackburn, and Mary A. Whooley. 2010. "Telomere Length Trajectory and Its Determinants in Persons with Coronary Artery Disease: Longitudinal Findings from the Heart and Soul Study." *PLoS ONE* 5 (1): e8612. <https://doi.org/10.1371/journal.pone.0008612>.
- Huzen, J, L S M Wong, D J Van Veldhuisen, N J Samani, A H Zwinderman, V Codd, R M Cawthon, et al. 2014. "Telomere Length Loss Due to Smoking and Metabolic Traits." *Journal of Internal Medicine* 275: 155–63. <https://doi.org/10.1111/joim.12149>.
- Kim, Wanil, Andrew T. Ludlow, Jaewon Min, Jerome D. Robin, Guido Stadler, Ilgen Mender, Tsung Po Lai, Ning Zhang, Woodring E. Wright, and Jerry W. Shay. 2016. "Regulation of the Human Telomerase Gene TERT by Telomere Position Effect-over Long Distances (TPE-OLD): Implications for Aging and Cancer." *PLoS Biology* 14 (12): 1–25. <https://doi.org/10.1371/journal.pbio.2000016>.
- Müezzinler, Aysel, Ute Mons, Aida Karina, Katja Butterbach, Kai-uwe Saum, Matthias Schick, Hermann Stammer, et al. 2015. "Smoking Habits and Leukocyte Telomere Length Dynamics among Older Adults: Results from the ESTHER Cohort." *Experimental Gerontology* 70: 18–25. <https://doi.org/10.1016/j.exger.2015.07.002>.
- Needham, Belinda L, Xu Wang, Judith E Carroll, Sharrelle Barber, Brisa N Sánchez, Teresa E Seeman, and Ana V Diez. 2019. "Sociodemographic Correlates of Change in Leukocyte Telomere Length during Mid- to Late-Life: The Multi-Ethnic Study of Atherosclerosis." *Psychoneuroendocrinology* 102: 182–88. <https://doi.org/10.1016/j.psyneuen.2018.12.007>.
- Nettle, D., L.A. Seeker, D. Nussey, H. Froy, and M. Bateson. 2018. "Consequences of Measurement

Error in QPCR Telomere Data: A Simulation Study.” *BioRxiv*, doi.org/10.1101/491944.

Révész, Dóra, Yuri Milaneschi, Erik M Terpstra, and Brenda W J H Penninx. 2016. “Baseline Biopsychosocial Determinants of Telomere Length and 6-Year Attrition Rate.” *Psychoneuroendocrinology* 67: 153–62. <https://doi.org/10.1016/j.psyneuen.2016.02.007>.

Robin, Jerome D., Andrew T. Ludlow, Kimberly Batten, Frederique Magdinier, Guido Stadler, Kathyrin R. Wagner, Jerry W. Shay, and Woodring E. Wright. 2014. “Telomere Position Effect: Regulation of Gene Expression with Progressive Telomere Shortening over Long Distances.” *Genes and Development* 28 (22): 2464–76. <https://doi.org/10.1101/gad.251041.114>.

Steenstrup, Troels, Jacob V B Hjelmborg, Jeremy D. Kark, Kaare Christensen, and Abraham Aviv. 2013. “The Telomere Lengthening Conundrum - Artifact or Biology?” *Nucleic Acids Research* 41 (13): 1–7. <https://doi.org/10.1093/nar/gkt370>.

Toupance, Simon, Carlos Labat, Mohamed Temmar, Patrick Rossignol, Masayuki Kimura, Abraham Aviv, Athanase Benetos, and See Editorial Commentary. 2017. “Short Telomeres, but Not Telomere Attrition Rates, Are Associated with Carotid Atherosclerosis.” *Hypertension* 70: 420–26. <https://doi.org/10.1161/short>.

Weischer, Maren, Stig E. Bojesen, and Børge G. Nordestgaard. 2014. “Telomere Shortening Unrelated to Smoking, Body Weight, Physical Activity, and Alcohol Intake: 4,576 General Population Individuals with Repeat Measurements 10 Years Apart.” *PLoS Genetics* 10 (3). <https://doi.org/10.1371/journal.pgen.1004191>.

Appendix B

Response to referees round 2

Reviewer: 2

Comments to the Author(s)

I appreciate the authors' response to my comments. The manuscript has been revised in accordance with my suggestions. I believe the findings support the authors' conclusions and can contribute significantly to the literature on the effects of smoking on telomere attrition.

I have one remaining suggestion. While re-reading and reflecting on the manuscript I realized that perhaps the authors can begin and end their manuscript by pointing to the overall damaging effects of smoking on health. While they have effectively shown that smoking does not accelerate the rate of telomere attrition and that selective adoption is a plausible explanation, it should be clarified that their results do not preclude the myriad of other damaging effects of smoking that are well established by the literature. I mention this because while I trust the author's themselves have no conflict of interest, ethically it is key that they help mitigate potential exploitative uses of their findings by the tobacco industry and its stakeholders.

Response

We completely understand the concerns of this referee and have no wish for our results to be misinterpreted as evidence against the harmful effects of smoking. We have therefore inserted sentences into the introduction and discussion to clarify our position, as suggested. We are also working on a press release to accompany the article that we hope will mitigate any misinterpretation.

We have added the following sentence to the second paragraph of the introduction: 'Smoking undoubtedly has a myriad of negative effects on human health and longevity. Moreover, the hypothesis that smoking causes telomere attrition is mechanistically plausible....'

We have also added the following final paragraph to the discussion: 'As a final note, although we found no evidence that smoking accelerates the rate of leukocyte telomere attrition, our results do not preclude the many other well-established negative effects of smoking on human health and longevity. We chose to focus on smoking in the current paper simply because there are more data available on the associations between smoking and telomere length than for any other behaviour. Our intention was to question prevalent assumptions in the telomere dynamics literature concerning the mechanisms underlying associations between behaviour and telomere length, rather than to question the damaging effects of smoking.'